# SARS-CoV-2 ORF7a potently inhibits the antiviral effect of the host factor SERINC5

Uddhav Timilsina[1], Supawadee Umthong[1], Emily B. Ivey [1], Brandon Waxman[1] & Spyridon Stavrou [1✉]

Serine Incorporator 5 (SERINC5), a cellular multipass transmembrane protein that is involved in sphingolipid and phosphatydilserine biogenesis, potently restricts a number of retroviruses, including Human Immunodeficiency Virus (HIV). SERINC5 is incorporated in the budding virions leading to the inhibition of virus infectivity. In turn, retroviruses, including HIV, encode factors that counteract the antiviral effect of SERINC5. While SERINC5 has been well studied in retroviruses, little is known about its role in other viral families. Due to the paucity of information regarding host factors targeting Severe Acute Respiratory Syndrome Coronavirus 2 (SARS-CoV-2), we evaluated the effect of SERINC proteins on SARS-CoV-2 infection. Here, we show SERINC5 inhibits SARS-CoV-2 entry by blocking virus-cell fusion, and SARS-CoV-2 ORF7a counteracts the antiviral effect of SERINC5 by blocking the incorporation of over expressed SERINC5 in budding virions.

---

[1] Department of Microbiology and Immunology, Jacobs School of Medicine and Biomedical Sciences, University at Buffalo, Buffalo, NY, USA.
✉email: stavrou2@buffalo.edu

n 2019, a novel pathogen associated with severe pneumonia was discovered and later identified to be a coronavirus, which was subsequently named Severe Acute Respiratory Syndrome Coronavirus 2 (SARS-CoV-2)[1] and is responsible for the current pandemic. Disease hallmarks range from mild to severe, including loss of smell and taste, breathing difficulty, fever, malaise and in elderly patients or people with preexisting illnesses, death[2].

An important determinant of virus infectivity is cellular entry[3,4]. Coronaviruses enter a target cell by first binding to a cell surface receptor followed by endosomal entry and fusion between the host and viral membranes[5]. In the case of SARS-CoV-2, the Spike (S) protein is critical for virus entry into human cells[5]. SARS-CoV-2 S is a transmembrane protein that forms a homotrimer on the surface of mature virions[6]. The S protein contains two subunits, S1 and S2. S1 interacts via its Receptor Binding Domain with the human angiotensin-converting enzyme 2 (ACE2) receptor on the surface of cells[5,7], and S2 facilitates fusion between virus and target cell membrane. SARS-CoV-2 S contains a furin cleavage site at the S1/2 boundary that is processed in the producer cell during virus production[6]. Subsequent activation of SARS-CoV-2 S by the target cell proteases cathepsin B/L and type II membrane serine protease 2 (TMPRSS2) is critical for virus entry into the cell, as it permits fusion between the viral and cellular membranes[8,9].

Viral entry, including that of SARS-CoV-2, is a major target for the development of intervention strategies[10,11]. However, the cellular host factors that regulate SARS-CoV-2 entry are not fully understood. Members of the Serine Incorporator (SERINC) protein family, which are thought to play an important role in sphingolipid and phosphatidylserine biogenesis[12], have important antiviral functions against retroviruses including Human Immunodeficiency Virus 1 (HIV-1) and Murine Leukemia Virus (MLV)[13,14]. SERINC5, and to a lesser extent SERINC3, are incorporated in the budding virions and interfere with the fusion of the viral and cellular membranes[15,16]. In turn, HIV-1 encodes a viral protein that counteracts the antiviral effect of this host factor. Negative effector factor (Nef) counteracts SERINC5 by removing it from the plasma membrane and preventing SERINC5 incorporation in the budding virions[13,14]. Furthermore, SERINC5 restricts retrovirus infection in vivo[17], which further highlights the importance of this host factor in virus restriction.

In this work, we elucidate the role of SERINC proteins in SARS-CoV-2 infection. We find that SERINC5 is expressed in pneumocytes and potently inhibits SARS-CoV-2 infection by binding to SARS-CoV-2 S, thereby blocking the fusion step during virus entry. In addition, we demonstrate that SARS-CoV-2 ORF7a, a SARS-CoV-2 accessory protein, counteracts the antiviral effect of SERINC5. SARS-CoV-2 ORF7a counteracts SERINC5 by utilizing two distinct mechanisms; it blocks SERINC5 incorporation in budding virions and is incorporated in budding virions, forming a complex with SERINC5 and SARS-CoV-2 S as a means of blocking SERINC5-mediated restriction of virus infectivity. In summary, we determine the mechanism SERINC5 employs to block SARS-CoV-2 infection, and we identify a previously undescribed function of SARS-CoV-2 ORF7a that is conserved among coronaviruses.

## Results

**Transcriptional regulation of *SERINC* genes**. To elucidate the role of *SERINC* genes during SARS-CoV-2 infection, we initially examined the levels of all the members of the SERINC protein family (SERINC1-5) in total lung tissue RNA from three donors and Calu-3 cells, a human pneumocyte cell line, as pneumocytes are natural targets of SARS-CoV-2 infection[18]. We performed RT-qPCR to determine the expression levels of the different

members of the SERINC family and found that all *SERINC* genes were expressed in human lungs and Calu-3 cells except for *SERINC4* (Fig. 1a, b). Thus, we excluded *SERINC4* from further studies. *ACE2* is an interferon stimulated gene (ISG) in lung epithelial cells, yet transcriptomic data from IFN treated peripheral blood mononuclear cells (PBMCs) never characterized *ACE2* as an ISG[19]. Similarly, *SERINC* gene expression is not induced by type I IFN in PBMCs[13]. Therefore, we sought to determine if IFN-β upregulates *SERINC* gene transcription in Calu-3 cells. We treated Calu-3 cells with IFN-β (500 U/ml), harvested cells at different time points, and isolated RNA followed by RT-qPCR to determine changes in *SERINC* gene expression. We found that the transcription levels of all *SERINC* genes examined (*SERINC1, 2, 3, and 5*) were unaffected by the presence of IFN-β in Calu-3 cells (Fig. 1c). We also determined the effect of SARS-CoV-2 infection on the expression levels of *SERINC* genes. We infected Calu-3 cells with SARS-CoV-2 (5 MOI) (USA-WA1/2020[20]) and harvested cells 4 and 6 h post infection (hpi). RNA was isolated and RT-qPCR was performed to determine viral RNA levels and possible changes in the expression levels of *SERINC1, 2, 3, and 5*. We found that SARS-CoV-2 infection had no effect on the transcription levels of *SERINC* genes (Fig. 1d, e). Therefore, we concluded that IFN-β and SARS-CoV-2 infection do not affect *SERINC* genes expression levels.

**SERINC5 potently restricts SARS-CoV-2 S-mediated entry**. SERINC5 exerts a potent antiretroviral effect by restricting HIV-1 entry, thereby decreasing virion infectivity[13,14]. We initially examined the role of SERINC proteins on SARS-CoV-2 entry by generating one-hit luciferase reporter SARS-CoV-2 S pseudoviruses using a replication-defective HIV-1 proviral plasmid (pHIV-1$_{NL}$ΔEnv-NanoLuc[21]) in the presence of SERINC1, 2, 3 and 5. To determine if SERINC proteins are incorporated in SARS-CoV-2 S pseudoviruses, we concentrated SARS-CoV-2 S pseudoviruses by ultracentrifugation from the media of the transfected cells and performed western blots. We found that all SERINC proteins examined (SERINC1, 2, 3, and 5) are packaged in SARS-CoV-2 S pseudovirions (Fig. 2a). We then infected Calu-3 and 293T-hACE2 cells using SARS-CoV-2 S pseudoviruses produced in the presence of the different *SERINC* genes and found that the presence of SERINC5 in SARS-CoV-2 S pseudoviruses resulted in a significant reduction in virus infectivity in both Calu-3 and 293T-hACE2 cells (Fig. 2b). SERINC3 reduced virus infectivity only in Calu-3 cells, albeit not as robustly as SERINC5, while SERINC1 and 2 had no effect (Fig. 2b). Thus, we concluded that SERINC5 blocks SARS-CoV-2 entry. To ensure that the SERINC5-mediated reduction in SARS-CoV-2 S-mediated entry is not due to the HIV-1 proviral packaging plasmid we used, we produced SARS-CoV-2 S pseudoviruses in the presence of SERINC3 and 5 using an MLV packaging plasmid and a luciferase reporter followed by infection of 293T-hACE2 cells. As a positive control, we used amphotropic MLV envelope pseudovirions, which are known to be restricted by SERINC5[17,22]. Similar to our studies with the lentiviral packaging system, we found that SARS-CoV-2 S pseudovirions produced in the presence of SERINC5 and the MLV packaging plasmid were significantly less infectious than those produced in the presence of empty vector (E.V.) (Supplementary Fig. 1), while SERINC3 had only a modest effect on virus infectivity (Supplementary Fig. 1). Thus, our results show that SERINC5, and to a lesser extent SERINC3, block SARS-CoV-2 S-mediated entry regardless of the packaging system we used.

While we clearly see that SERINC5 blocks SARS-CoV-2 S-mediated entry and is incorporated in SARS-CoV-2 S pseudoviruses, this does not necessarily reflect what happens with coronaviruses. Pseudoviruses generated using a lentiviral packaging

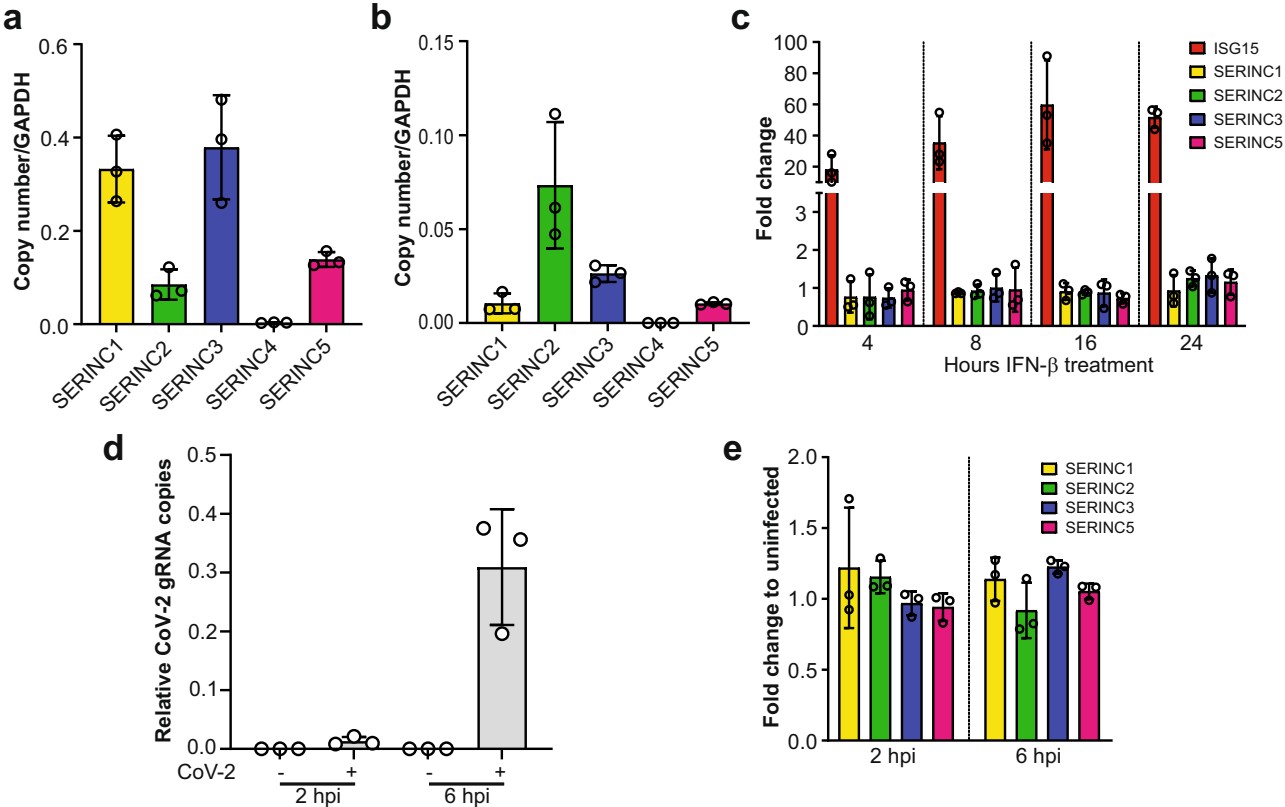

**Fig. 1 *SERINC* genes are expressed in pneumocytes and their expression levels are unaffected by SARS-CoV-2 infection and type I IFN. a** SERINC1, 2, 3, 4, and 5 RNA copy number relative to GAPDH in total lung RNA. Graphs represent mean ± SD from three independent donors. **b** SERINC1, 2, 3, 4, and 5 RNA copy number relative to GAPDH in Calu-3 cells. **c** Fold expression change of SERINC1, 2, 3, 5, and ISG15 transcripts relative to mock, normalized to GAPDH in Calu-3 cells treated with IFNβ (500 U/ml) for 4 h, 8 h, 16 h or 24 h. Mock indicates mock-treated (PBS). **d** SARS CoV-2 gRNA copies relative to GAPDH from Calu-3 cells infected with SARS-CoV-2 for 2 h or 6 h **e** SERINC1, 2, 3, and 5 transcripts fold expression from **d** relative to uninfected Calu-3 cells and normalized to GAPDH. All graphs represent mean ± SD from 3 independent experiments. (hours post infection, hpi; SARS-CoV-2, CoV-2).

plasmid, assemble at the plasma membrane[23]. On the other hand, coronaviruses, including SARS-CoV-2, assemble at the endoplasmatic reticulum Golgi intermediated compartment (ERGIC)[24]. Therefore, SERINC5 incorporation in pseudoviruses does not necessarily equate with SERINC5 incorporation in SARS-CoV-2 virions. To ascertain that SERINC5 is incorporated in SARS-CoV-2 virions, we employed two methodologies, using both virus-like particles (VLPs) and infectious virus. We initially evaluated SERINC5 incorporation in VLPs; VLPs generated by transfecting all SARS-CoV-2 structural proteins self-assemble at the ERGIC, similar to infectious SARS-CoV-2 virions[25,26]. Consequently, VLPs are an accurate model to study SARS-CoV-2 assembly. We co-transfected 293T cells with plasmids encoding SARS-CoV-2 N (Nucleoprotein), M (Membrane), E (Envelope), and S in the presence or absence of SERINC5. At 48 h post transfection, cells were lysed and VLPs in the culture media of the transfected cells were concentrated by ultracentrifugation. We then performed western blots and found that SERINC5 was incorporated in the purified VLPs (Fig. 2c). Therefore, we concluded that SERINC5 is incorporated in SARS-CoV-2 VLPs during assembly at the ERGIC. To determine whether SERINC3 and 5 are incorporated in infectious SARS-CoV-2 virions, 293T-hACE2 expressing either SERINC5 or SERINC3 were infected with SARS-CoV-2 (USA-WA1/2020[20]). We purified virus from the culture media of the cells and performed western blots. We observed that SERINC5 was incorporated quite efficiently in SARS-CoV-2 virions, while SERINC3 was incorporated at lower levels (Fig. 2d). Unfortunately, we could not examine incorporation of endogenous SERINC3 and

SERINC5 levels as the antibodies available are not suitable for western blots.

To determine the effect of incorporated SERINC3 and 5 on infectious SARS-CoV-2, we infected 293T-hACE2 and Calu-3 cells with equal amounts of SARS-CoV-2 (USA-WA1/2020[20]) (normalized for RNA levels) isolated from the media of infected 293T-hACE2 cells expressing either E.V., SERINC3 or SERINC5. At 6 hpi RNA was isolated from the infected cells followed by RT-qPCR. We found that SERINC5 reduced SARS-CoV-2 infectivity, while SERINC3 had no effect (Fig. 2e). To ensure that the RNA levels detected were due to infection and not input virus inoculum, we also infected *Mus dunni* cells (mouse fibroblasts) with SARS-CoV-2, as mouse cells do not get infected by SARS-CoV-2 and any viral RNA detected will be from the input inoculum. As expected, very low levels of viral RNA were detected (Fig. 2e). In conclusion, our data using pseudoviruses, VLPs, and infectious SARS-CoV-2 demonstrate that SERINC5 restricts SARS-CoV-2 infection by blocking virus entry. Therefore, in this report we have focused on the role of SERINC5 on SARS-CoV-2 infection.

**SERINC5 blocks SARS-CoV-2 entry of different Spike variants.** It was previously shown that envelopes derived from different HIV-1 strains differ in their sensitivity to SERINC5-mediated restriction[16]. During the pandemic spread of SARS-CoV-2, a number of variants have emerged, which are associated with high transmissibility[27,28]. The SARS-CoV-2 variants B.1.1.7 (Alpha

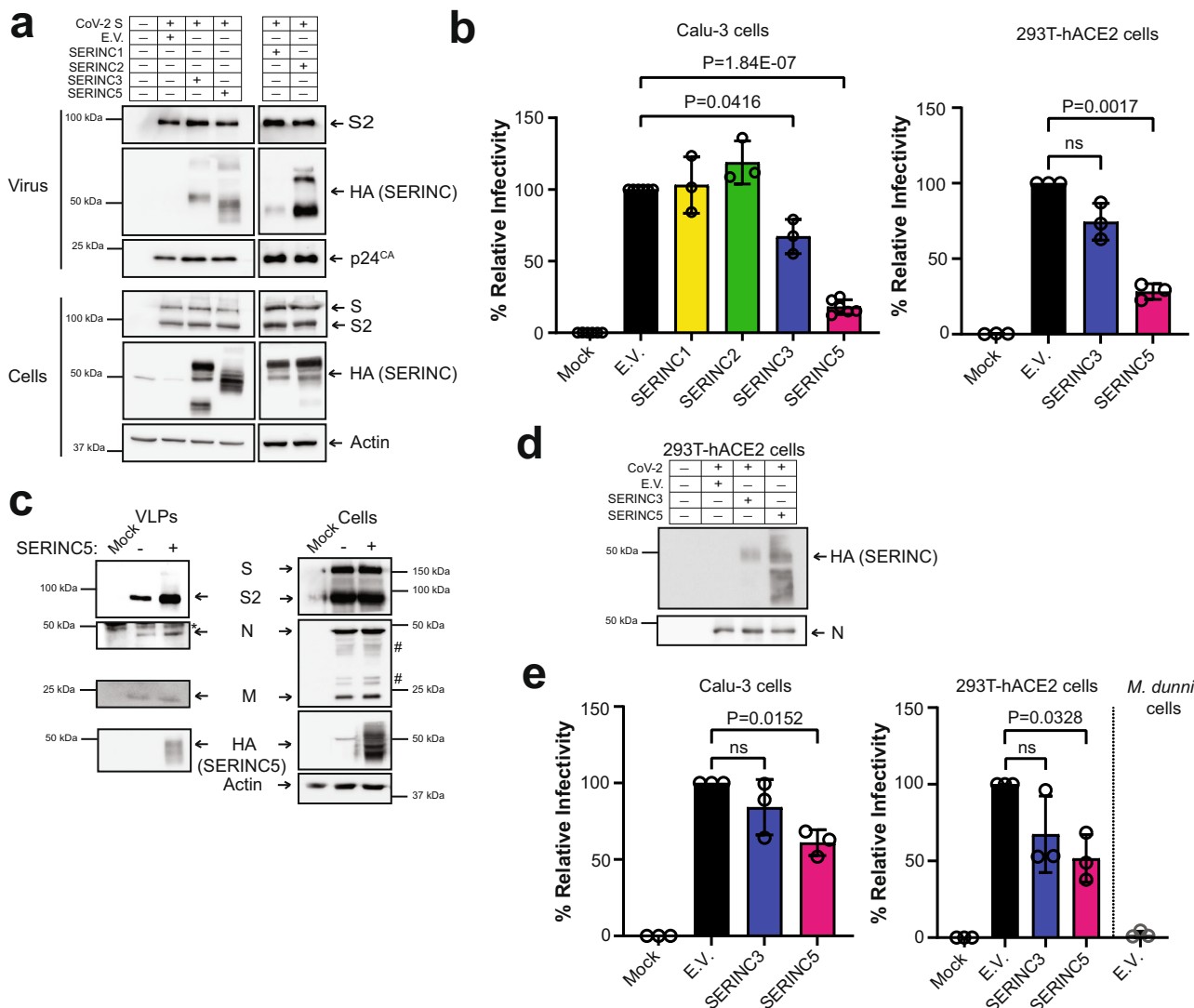

**Fig. 2 SERINC proteins are incorporated into SARS-CoV-2 virus and restrict SARS-CoV-2 S-mediated entry. a** SERINC proteins are incorporated in SARS-CoV-2 S pseudoviruses. 293T cells were co-transfected with HIV-1$_{NL}$ΔEnv-NanoLuc, SARS-CoV-2 Spike and either SERINC1, 2, 3, 5, or empty vector (E.V.) as indicated. Subsequently, cells and released pseudovirus in the culture media were harvested and the indicated proteins were analyzed by immunoblotting. **b** SERINC5 blocks SARS-CoV-2 S–mediated entry. Calu-3 or 293T-hACE2 cells were infected with SARS-CoV-2 S pseudoviruses produced in the presence of SERINC1, 2, 3, 5, or E.V. Luciferase levels were measured 48 hpi and normalized to HIV-1 p24$^{CA}$ levels of the input virus. The percentage (%) of relative infectivity with respect to pseudovirus produced in the presence of empty vector is shown. **c** SERINC5 is incorporated in virus-like particles (VLPs). VLPs were generated either with SERINC5 or E.V and subsequently analyzed by immunoblotting. * denotes nonspecific band, # represents glycosylated forms of M protein. **d** SERINC5 is incorporated in infectious SARS-CoV-2 virions. SARS-CoV-2 particles were produced in 293T-hACE2 cells expressing either SERINC3, SERINC5 or E.V. Virions released in the culture media were harvested and the indicated proteins were analyzed by immunoblotting. **e** SERINC5 inhibits SARS-CoV-2 infectivity. Calu-3, 293T-hACE2 or *Mus dunni* (*M. dunni*) cells were infected with SARS-CoV-2 virions (normalized for viral RNA copies) produced in the presence of SERINC3, 5 or E.V (from Fig. 2d). Six hours post infection, SARS-CoV-2 Spike RNA copy number in the infected cells were determined by RT-qPCR, normalized to GAPDH, and presented as % relative to E.V., which was set to 100%. For **a**, **c**, and **d**, representative immunoblotting results from *n* = 3 independent experiments are shown. Uncropped blots are in Source Data. For **b** and **e**, results are presented as mean ± SD from at *n* = 3 or 6 independent experiments. Statistical significance was determined by one-sample t-test (two-tailed). (empty vector, E.V.; SARS-CoV-2 Spike, S; SARS-CoV-2 Spike S2 subunit, S2; M, membrane protein; N, nucleocapsid protein; *Mus dunni* cells, *M. dunni* cells; not significant, ns).

variant), B.1.351 (Beta variant), P.1 (Gamma variant) and B.1.617 (Delta variant) are highly transmissible and their mutations in the S protein are responsible for the increased transmissibility[27,29,30]. Therefore, it is possible that S proteins derived from different SARS-CoV-2 variants may differ in their susceptibility to SER-INC5 restriction. To determine the effect of SERINC5 on S proteins derived from different SARS-CoV-2 variants, we generated one-hit luciferase reporter SARS-CoV-2 S pseudoviruses in

the presence or absence of SERINC5, using a replication-defective HIV-1 proviral plasmid described above and SARS-CoV-2 S derived from the Alpha (B.1.1.7), Beta (B.1.351), Gamma (P1), and Delta (B.1.617) variants respectively. Afterwards, Calu-3 cells were infected and luciferase levels were measured 48 hpi. We found that SERINC5 restricted infection of all viral particles regardless of the SARS-CoV-2 variant (Alpha, Beta, Gamma and Delta) from which the S protein was derived (Fig. 3). In

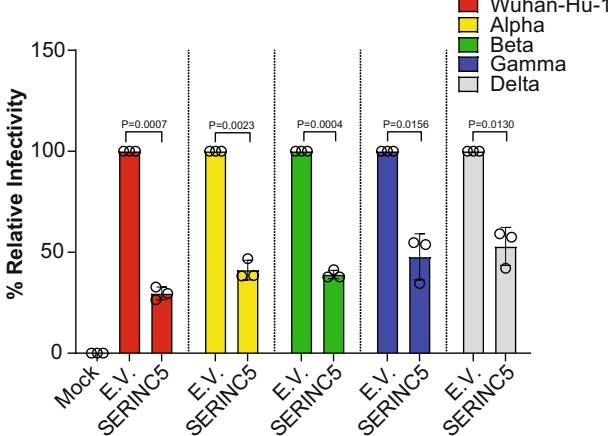

**Fig. 3 SERINC5 restricts SARS-CoV-2 entry of different Spike variants.** 293T cells were co-transfected with HIV-1$_{NL}$ΔEnv-NanoLuc, SARS-CoV-2 Spike variants (Wuhan-Hu-1, alpha, beta, gamma, and delta) and either SERINC5 or empty vector (E.V.). Forty-eight hours post transfection, released virus in the culture media was used to infect Calu-3 cells. Luciferase levels were measured 48 hpi and normalized to HIV-1 p24$^{CA}$ levels of the input pseudovirus. Percent (%) of relative infectivity with respect to pseudovirus produced in the presence of E.V. is shown. All results are presented as mean ± SD from $n = 3$ independent experiments. Statistical significance was determined by one-sample t-test (two-tailed).

conclusion, our data show that SERINC5 restriction of SARS-CoV-2 S-mediated entry is consistent across the different SARS-CoV-2 variants.

**SERINC5 inhibits SARS-CoV-2 by targeting virus-cell membrane fusion.** SARS-CoV-2 entry into the cell is a complex process of events that involves the coordination of receptor interaction and proteolytic cleavages of the S protein that culminates with virus-cell membrane fusion. To determine which step of SARS-CoV-2 S-mediated entry is affected by SERINC5, we initially examined the effect of SERINC5 on SARS-CoV-2 S binding to the ACE2 receptor. 293T-hACE2 cells were incubated on ice for 1 h with FITC-labeled SARS-CoV-2 S pseudovirions produced in the presence or absence of SERINC5. Subsequently, cells were shifted for 1 h at 37 °C and analyzed by flow cytometry. We found that SERINC5 incorporation did not affect the amount of virus bound to 293T-hACE2 cells (Fig. 4a). As negative control, we used 293T cells that did not express the ACE2 receptor and as expected we saw only minimal amounts of virus bound to the cells (Fig. 4a). Thus, we concluded that SERINC5 does not influence the SARS-CoV-2 S-ACE2 interaction.

SARS-CoV-2 S activation by cellular proteases is an important step of virus entry[31]. Cathepsins found in the lysosomes and endosomes are critical for SARS-CoV-2 S activation and infection of 293T-hACE2 cells[5,31]. In order to facilitate SARS-CoV-2 fusion with the plasma membrane of the target cell, SARS-CoV-2 S is initially cleaved in the producer cells by furin at the S1/S2 cleavage site. This is followed by cleavage at the S2' site, which is located proximally to the S1/2 cleavage site, by cathepsins in endosomal compartments or by TMPRSS2 on the surface of target cell[8,31,32]. Previous reports have shown that cathepsin inhibitors potently reduce SARS-CoV-2 entry[5,33]. Thus, it is possible that SERINC5 blocks SARS-CoV-2 entry by interfering with the protease-mediated activation at S2' of SARS-CoV-2 S. To address this, we used a previously described SARS-CoV-2 S, in which the furin cleavage site has been mutated and thus cleavage only occurs by either cathepsins or TMPRSS2 at the S2' site

(SARS-CoV-2 S-FKO –Furin KnockOut)[34]. We co-transfected 293T cells with SARS-CoV-2 S-FKO, the lentiviral packaging plasmid and either SERINC5 or E.V. At 48 h post transfection, pseudovirions produced in the presence of SERINC5 or E.V. were concentrated by ultracentrifugation and verified by western blot for SERINC5 incorporation (Supplementary Fig. 2) followed by incubation with cathepsin L (10 μg/ml) as previously performed[35]. For our assay, we used a cathepsin L inhibitor (SID 26681509) as a negative control. Western blots were then performed to examine cathepsin L-mediated cleavage at S2' of SARS-CoV-2 S-FKO in the presence or absence of SERINC5. We found that SARS-CoV-2 S-FKO was cleaved by cathepsin L at similar levels both in the presence of SERINC5 or E.V. (Fig. 4b). As expected, the use of a cathepsin L inhibitor decreased the levels of cleaved SARS-CoV-2S-FKO in the presence of cathepsin L (Fig. 4b). Thus, we concluded that SERINC5 has no effect on cathepsin L-mediated cleavage of SARS-CoV-2 S and does not interfere with the proteolytic activation of SARS-CoV-2 S.

As SERINC5 does not interfere with receptor binding or the proteolytic activation of SARS-CoV-2 S, we next investigated if SERINC5 interferes with the fusion step between the virus and the target cell membrane. To address the role of SERINC5 on SARS-CoV-2 S-mediated fusion, we took advantage of the sensitive beta-lactamase-Vpr (BlaM-Vpr) fusion assay, which involves packaging of BlaM-Vpr chimeric protein in virions followed by its delivery into the cytoplasm of target cells as a result of virus-cell fusion, and has been previously used to study coronavirus fusion[36]. The transfer of BlaM-Vpr to the cytosol of the target cell is detected by enzymatic cleavage of the CCF2 dye, a fluorescent substrate of BlaM, loaded in the target cells[37]. We generated BLaM-Vpr containing SARS-CoV-2 S pseudoviruses in the presence of either SERINC5 or E.V. Following normalization for p24$^{CA}$, 293T-hACE2 cells were infected with equal amounts of SARS-CoV-2 S pseudoviruses in the presence or absence of a neutralizing anti-SARS-CoV-2 S antibody and processed by flow cytometry. We found that the SARS-CoV-2 S-mediated fusion was decreased in the presence of SERINC5 (Fig. 4c). The addition of a neutralizing antibody abrogated SARS-CoV-2 S-mediated fusion (Fig. 4c). Therefore, we conclude that SERINC5 blocks SARS-CoV-2 entry by inhibiting SARS-CoV-2 S-mediated fusion.

**SARS-CoV-2 ORF7a counteracts the antiviral function of SERINC5.** Many retroviruses encode factors (HIV-1 Nef) that counteract SERINC5 by blocking its incorporation in budding virions[14,22,38]. However, viral antagonists of SERINC5 from other viral families have not been identified. Consequently, we examined the SARS-CoV-2 accessory proteins, as they are implicated in blocking host antiviral genes and are important for virus pathogenesis[39,40]. For our work we focused on ORF7a, a type I transmembrane protein of 121 amino acids that is very similar (85% sequence identity) with SARS-CoV ORF7a[40]. SARS-CoV ORF7a is found in the ER and Golgi as well as in purified virions[41]. Therefore, we hypothesized that SARS-CoV-2 ORF7a may be interfering with the antiviral function of SERINC5. To determine the effect of SARS-CoV-2 ORF7a on SERINC5-mediated restriction of SARS-CoV-2 entry, we co-transfected 293T cells with the aforementioned HIV-1 proviral replication defective plasmid, SARS-CoV-2 S and SERINC5 along with different amounts of SARS-CoV-2 ORF7a. Virus was concentrated by ultracentrifugation from the media of the transfected cells and western blots were performed to verify the presence of virus (Supplementary Fig. 3a). Viruses were used to infect Calu-3 and 293T-hACE2. At 48 hpi, we measured luciferase levels and found that SARS-CoV-2 ORF7a alleviated the antiviral effect of SERINC5 on virus infectivity in a dose-dependent manner in Calu-3

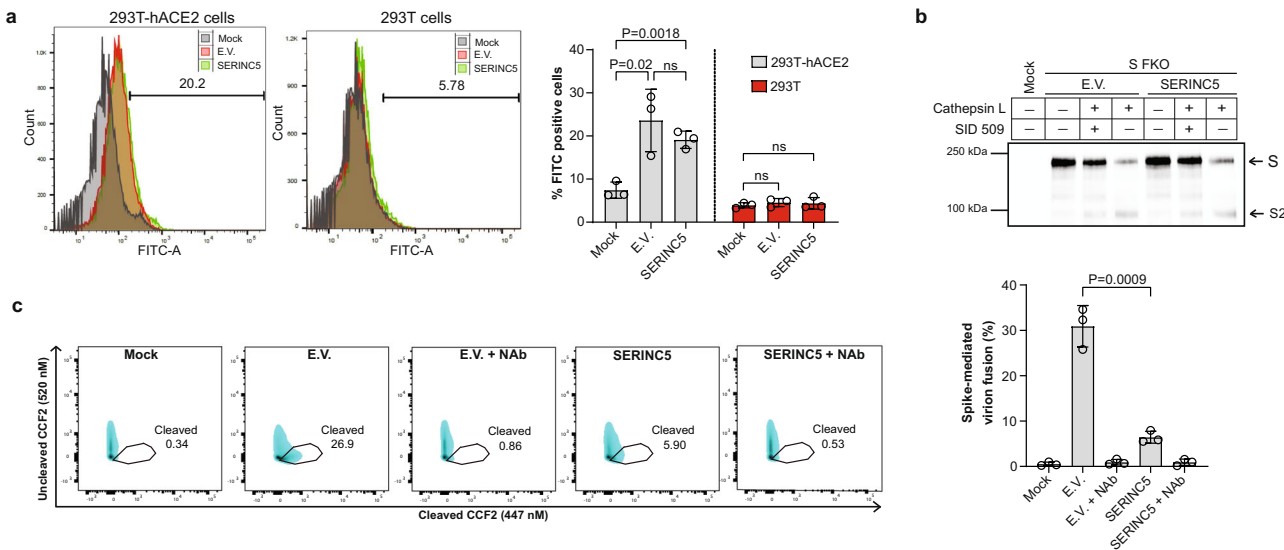

**Fig. 4 SERINC5 blocks SARS-CoV-2 entry by targeting virus-cell membrane fusion. a** SERINC5 does not affect SARS-CoV-2 S binding to ACE2 receptor. 293T-hACE2 or 293T cells were incubated on ice for 1 h with FITC-labeled SARS-CoV-2 S pseudovirions produced in the presence of either SERINC5 or empty vector (E.V.), shifted to 37 °C for 1 h and analyzed by FACS. Shown on the right are representative FACS plots and the graph on the left represents mean ± SD of % FITC positive cells from *n* = 3 independent experiments. Statistical significance was determined by unpaired t-test (two-tailed). 293T cells served as negative controls. **b** SERINC5 does not affect SARS-CoV-2 S priming by cathepsin L. SARS-CoV-2 S-FKO pseudoviruses produced in the presence of either SERINC5 or E.V. were incubated with 10 µg/ml of cathepsin L for 1 h at RT with or without SID 26681509, a cathepsin L inhibitor. Samples were analyzed by immunoblotting using anti-FLAG antibody. Shown here is a representative immunoblot image from *n* = 3 independent experiments. Uncropped blots are in Source Data. **c** SERINC5 inhibits SARS-CoV-2 S-mediated fusion. 293T-hACE2 cells were infected with equal amounts of SARS-CoV-2 S pseudoviruses produced in the presence of either SERINC5 or E.V. Pseudoviruses pre-incubated with neutralizing SARS-CoV-2 S antibody were also included in these experiments. Cells were stained with CCF2-AM and then subjected to FACS. Fusion efficiency was quantified as the percentage of cleaved CCF2. Representative FACS plots are shown on the right. The bars on the left represent mean ± SD of % cleaved CCF2 positive cells from *n* = 3 independent experiments. Gating strategies are shown in Supplementary Fig. 7. Statistical significance was determined by unpaired t-test (two-tailed). (empty vector, E.V.; SARS-CoV-2 Spike, S; SARS-CoV-2 Spike S2 subunit, S2; SARS-CoV-2 Spike-FKO, S FKO; SID 26681509, SID 509; Neutralizing Antibody, Nab; not significant, ns).

cells and 293T-hACE2 cells (Fig. 5a). HIV Nef, a known viral antagonist of SERINC5, blocks the antiviral effect of SERINC5 by preventing its incorporation in nascent virions[13,14,17]. Therefore, we performed western blots on purified pseudovirions to determine the levels of SERINC5 incorporated. In the case of SARS-CoV-2 S pseudoviruses, SARS-CoV-2 ORF7a did not prevent the incorporation of SERINC5 in virions or affect the cellular levels of SERINC5 in the cell fractions of the transfected cells (Fig. 5b). Finally, similar to SARS-CoV ORF7a, we found that SARS-CoV-2 ORF7a can be incorporated in budding pseudoviruses (Fig. 5b).

We also examined the effect of ORF7a on SERINC5 restriction in the context of SARS-CoV-2 infection. For these studies we used wild type (WT) SARS-CoV-2 (USA-WA1/2020) and SARS-CoV-2-eGFP virus, in which the ORF7a ORF in WT SARS-CoV-2 has been deleted and replaced with the eGFP gene[42]. In agreement with previous findings, both viruses replicate similarly in 293T-hACE2 cells (Supplementary Fig. 3b)[43]. Initially, we generated SARS-CoV-2 (WT) and SARS-CoV-2-eGFP (ΔORF7a) in the presence or absence of SERINC5 by infecting 293T-hACE2 cells expressing either SERINC5 or E.V. To determine the effect of ORF7a on the incorporation of SERINC5 in infectious SARS-CoV-2, we performed western blots on purified WT and ΔORF7a virions produced in the presence or absence of SERINC5. When evaluating the virus fraction, we noticed that ORF7a is packaged in budding virions (Fig. 5c). In addition, unlike our findings with SARS-CoV-2 S pseudoviruses, we found that deletion of ORF7a resulted in more SERINC5 packaged inside the nascent virions (Fig. 5c). Therefore, we concluded that in the context of SARS-CoV-2 infection ORF7a prevents the incorporation of SERINC5 in budding virions. To determine the effect of ORF7a on SARS-

CoV-2 infectivity, we performed a virus spread assay. 293T-hACE2 cells transfected with SERINC5 or E.V. were infected with WT or ΔORF7a virus. At 24, 36, and 48 hpi RNA was isolated from the infected cells followed by RT-qPCR. We found that compared to WT virus, replication of the ΔORF7a virus was reduced in the presence of SERINC5 (Fig. 5d). To further confirm the role of ORF7a-mediated counteraction of SERINC5 restriction, we complemented ORF7a in trans in infections with the ΔORF7a virus. 293T-hACE2 cells expressing SERINC5 were transfected with or without ORF7a followed by infection with the ΔORF7a virus. Virus containing culture supernatants and cells were collected 48 hpi followed by western blots. When looking at the virus fraction, we noticed that the addition of ORF7a in trans reduced the levels of SERINC5 incorporated in the budding ΔORF7a virions (Fig. 5e). Next, we used equal amounts of the purified virus (normalized for RNA levels) to infect 293T-hACE2 cells and infectivity was measured at 6 hpi by RT-qPCR. We found that *trans* complementation of ORF7a rescued the infectivity of ΔORF7a virus in the presence of SERINC5 (Fig. 5f). Thus, we conclude that ORF7a alleviates SERINC5 restriction of SARS-CoV-2.

Furthermore, we examined the effect of endogenous SERINC5 on SARS-CoV-2 particle infectivity by infecting cells with either SARS-CoV-2 WT or ΔORF7a produced in SERINC5-depleted cells. siRNA-mediated depletion of SERINC5 (siS5) was carried out in Calu-3 cells, which express endogenous SERINC5 (Fig. 1b); an siControl (siCon) served as a negative control. At 48 h after siRNA transfection, cells were infected with either WT or ΔORF7a virus. After knockdown verification (Supplementary Fig. 4a), viruses were collected 24 hpi and quantified by RT-

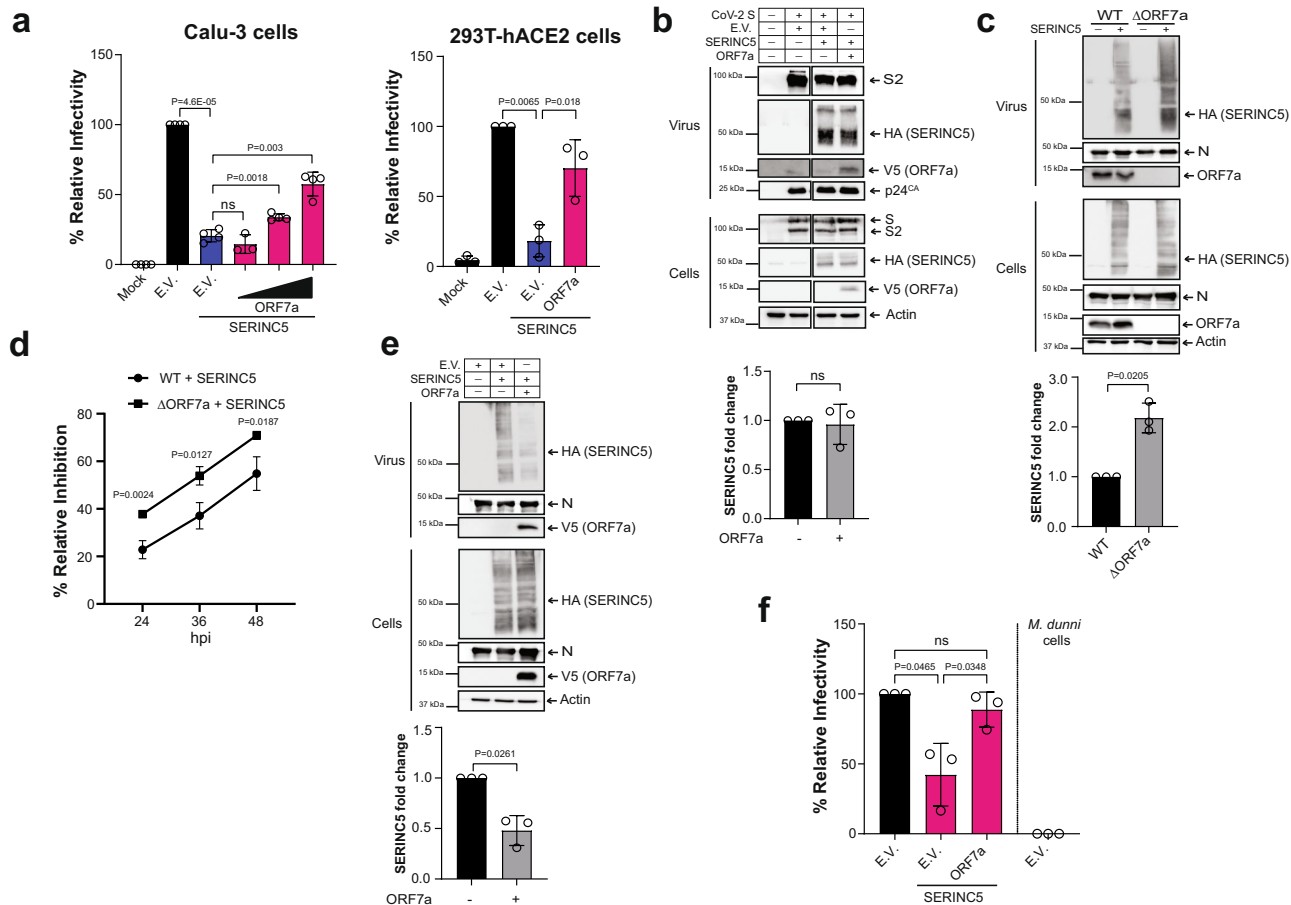

**Fig. 5 SARS-CoV-2 ORF7a counteracts SERINC5. a** SARS-CoV-2 ORF7a blocks SERINC5. Calu-3 cells and 293T-hACE2 cells were infected with SARS-CoV-2 S pseudoviruses produced with or without SERINC5 and ORF7a. Percentage (%) of relative infectivity with respect to pseudovirus produced with empty vector (E.V.) is shown. **b** ORF7a is incorporated in nascent virions and does not affect SERINC5 levels. SARS-CoV-2 S pseudovirions produced with SERINC5, ORF7a or E.V. were analyzed by immunoblotting. **c** ORF7a blocks the incorporation of SERINC5 in SARS-CoV-2 virions. SARS-CoV-2 WT and eGFP (ΔORF7a) viruses generated with either SERINC5 or E.V. were analyzed by immunoblotting. **d** SERINC5 restriction is countered by ORF7a. 293T-hACE2 cells expressing SERINC5 or E.V. were infected with SARS-CoV-2 WT or ΔORF7a virus. SARS-CoV-2 Spike RNA copy number in the infected cells were determined by RT-qPCR, normalized to GAPDH, and presented as the mean of % relative inhibition to E.V. set at each time point to 0%. ORF7a in trans rescues ΔORF7a virus infectivity when SERINC5 is present. **e** Cells and virus fractions of 293T-hACE2 cells expressing SERINC5, ORF7a or E.V. and infected with ΔORF7a virus were analyzed by immunoblotting. **f** Virus from **e** was used to infect 293T-hACE2 or *Mus dunni* cells. Six hpi, SARS-CoV-2 Spike RNA copy number in the infected cells were determined by RT-qPCR and % relative infectivity was determined with E.V. set to 100%. For **b**, **c**, and **e**, representative immunoblotting results are shown, bottom graphs represent densitometric analysis as mean ± SD for the fold change in virus-incorporated SERINC5 ± ORF7a from *n* = 3 independent experiments. Uncropped blots are in Source Data. For **a**, **d**, and **f**, data are presented as mean ± SD from *n* = 3 or 4 independent experiments. For **a**, **b**, **c**, **e**, and **f**, statistical significance was determined using one-sample t-test (two-tailed). Unpaired *t* test (two-tailed) was used in **d** and for **a** and **f** when non E.V. comparisons were performed. (hour post infection, hpi; empty vector, E.V.; SARS-CoV-2 Spike, S; SARS-CoV-2 Spike S2 subunit, S2; SARS-CoV-2 ORF7a, ORF7a; *Mus dunni* cells, *M. dunni* cells; not significant, ns).

qPCR. Calu-3 cells were infected with equal genome copies and harvested 6 hpi. To ensure that the viral RNA levels we detect at 6 h are not due to the inoculum, we infected *Mus dunni* cells in parallel. We performed RT-qPCR and found that viral RNA levels at 6 hpi were similar in Calu-3 cells infected with WT SARS-CoV-2 produced in cells treated with either siControl or siSERINC5 (Fig. 6a). On the other hand, viral RNA levels were increased in cells infected with ΔORF7a virions produced in cells depleted of SERINC5 compared to viral RNA levels from cells infected with ΔORF7a virions produced in cells treated with siControl (Fig. 6a). We also examined the effect of endogenous SERINC5 on SARS-CoV-2 spread using Calu-3 cells treated with either a SERINC5 (shS5-Calu-3) or Control (shCon-Calu-3) shRNA. After knockdown verification (Supplementary Fig. 4b), shS5-Calu-3 and shCon-Calu-3 cells were infected with equal

genome copies of SARS-CoV-2 WT or ΔORF7a viruses. RNA was isolated from the infected cells at 24 and 48 hpi followed by RT-qPCR analysis to measure the viral RNA levels. Unlike what we found in 293T-hACE2 cells (Supplementary Fig. 3b), SARS-CoV-2 ΔORF7a virus replicated at considerably lower levels than SARS-CoV-2 WT virus in Calu-3 cells (Supplementary Fig. 4c). Therefore, we have abstained from making any direct comparisons between these two viruses in Calu-3 cells. In the case of SARS-CoV-2 WT, we found that virus replicated similarly in both SERINC5 expressing (shCon-Calu-3) and SERINC5 depleted (shS5-Calu3) cells (Fig. 6b). On the other hand, SARS-CoV-2 ΔORF7a replicated at higher levels in SERINC5 depleted (sh5-Calu-3) cells compared to SERINC5 expressing (shCon-Calu-3) cells (Fig. 6c). Thus, we conclude that endogenous SERINC5 mitigates SARS-CoV-2 infection in the absence of ORF7a.

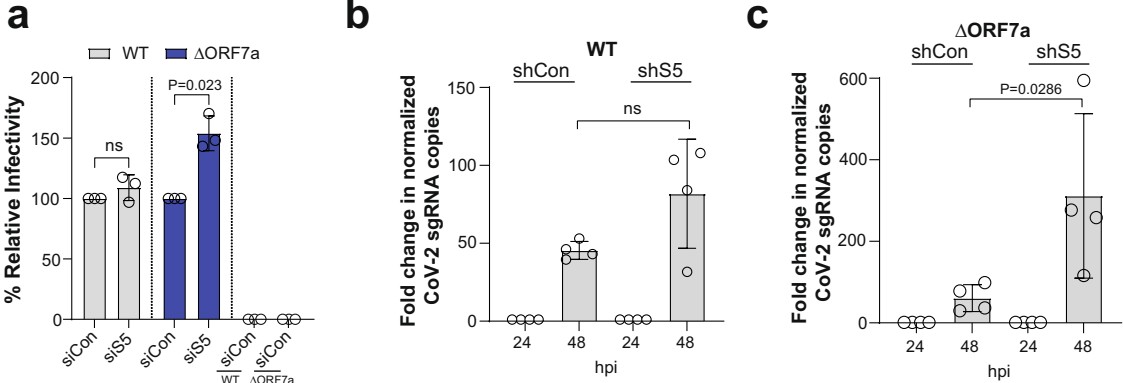

**Fig. 6 ORF7a counteracts endogenous SERINC5. a** Endogenous SERINC5 reduces SARS-CoV-2 entry in the absence of Orf7a. Calu-3 and *M. dunni* cells were infected with SARS-CoV-2 WT or ΔORF7a virus produced in Calu-3 cells treated with siSERINC5 or siControl. SARS-CoV-2 Spike RNA copy numbers were determined by RT-qPCR at 6 hpi and percentage (%) relative infectivity was determined by setting the infectivity of virus from siControl treated cells to 100%. Results are presented as mean ± SD from $n = 3$ independent experiments. One-sample t-test (two-tailed) was performed. Endogenous SERINC5 reduces SARS-CoV-2 replication in the absence of ORF7a. Calu-3 cells stably expressing shSERINC5 or shControl were infected with either **b** SARS-CoV-2 WT or **c** ΔORF7a virus. Cells were harvested at 24 and 48 hpi, SARS-CoV-2 Spike RNA copy numbers were determined by RT-qPCR and normalized to GAPDH. Graphs show fold change of normalized SARS-CoV-2 Spike RNA levels relative to 24 hpi conditions. Results are presented as mean ± SD from $n = 4$ independent experiments. Mann Whitney test (two-tailed) was performed. (hour post infection, hpi; negative control siRNA, siCon; SERINC5 siRNA, siS5; negative Control shRNA, shCon; SERINC5 shRNA, shS5; si, *Mus dunni* cells, *M. dunni* cells; not significant, ns).

Unfortunately, we could not verify at the protein level changes in endogenous SERINC5 incorporation in virions, as there is no antibody currently available for western blots.

**SARS-CoV-2 S, ORF7a, and SERINC5 co-localize at the ERGIC and form a complex.** SARS-CoV S and SARS-CoV ORF7a localize at the ERGIC[44,45]. Thus, we determined by immunofluorescence if SERINC5, SARS-CoV-2 S, and ORF7a also co-localize at the ERGIC. To address this, we co-transfected AD293 cells with expression plasmids for SARS-CoV-2 S, ORF7a, SERINC5 and ERGIC53-GFP, an ERGIC marker. Cells were fixed 24 h post transfection and stained with anti-HA (SERINC5), anti-Spike, and anti-V5 (ORF7a) followed by confocal microscopy. We found that SERINC5 (Spearman's rank correlation value 0.669), ORF7a (Spearman's rank correlation value 0.693) and SARS-CoV-2 S (Spearman's rank correlation value 0.814) co-localize at the ERGIC (Fig. 7a).

Our data show that ORF7a counteracts SERINC5 at two sites during infection, in the producer cells (by virion exclusion) and in the virions themselves. We hypothesized that in virions, ORF7a forms a complex with SERINC5 and SARS-CoV-2 S to block SERINC5 restriction of SARS-CoV-2 entry. To determine if SARS-CoV-2 S, ORF7a, and SERINC5 interact with one another, we co-transfected 293T cells with SARS-CoV-2 S, SERINC5, and SARS-CoV-2 ORF7a. We then performed coimmunoprecipitations (coIPs) with anti-SARS-CoV-2 S, anti-V5 (SARS-CoV-2 ORF7a) and anti-HA (SERINC5). We noticed that SARS-CoV-2 S coimmunoprecipitated with SARS-CoV-2 ORF7a and SERINC5 (Fig. 7b, lane 4), and SERINC5 coimmunoprecipitated with SARS-CoV-2 S and ORF7a (Fig. 7b, lane 10). We also detected SARS-CoV-2 S and SERINC5 in western blots when pulling down with SARS-CoV-2 ORF7a (Fig. 7b, lane 16). Furthermore, when we co-transfected SARS-CoV-2 S only with SERINC5 and performed coIPs with either anti-SARS-CoV-2 S or anti-HA (SERINC5), we found that these two proteins physically interact (Fig. 7b, lanes 3 and 9). Finally, previous reports showed SARS-CoV ORF7a and SARS-CoV S form a complex[41,44]. Our findings show that SARS-CoV-2 S and ORF7a also form a complex (Fig. 7b, lanes 5 and 17). To ensure that the interaction between ORF7a, Spike and SERINC5 is specific, we co-transfected

293T cells with ORF7a and TMED2, an ERGIC resident protein[46]. We performed co-IPs with anti-V5 (ORF7a) and immunoblots probing for FLAG (TMED2) and observed no interaction between ORF7a and TMED2 (Supplementary Fig. 5). In summary, our data show that ORF7a forms a complex with SERINC5 and SARS-CoV-2 S and the formation of this complex on the virion surface may block SERINC5 from restricting SARS-CoV-2 entry.

**Mapping the anti-SERINC5 domain of ORF7a.** SARS-CoV-2 ORF7a is a type I transmembrane protein of 121 amino acids[40]. ORF7a contains an immunoglobulin-like (Ig-like) ectodomain with 7 β-sheets (residues 16–96), a transmembrane domain (residues 97–116) and a short C' terminal tail (residues 117–121)[47]. Recent reports have identified a number of SARS-CoV-2 ORF7a deletions in viruses isolated from infected patients[48–50]. Naturally occurring deletions in ORF7a may affect its ability to counteract SERINC5. For our work we decided to focus on the deletions that did not affect the signal peptide, transmembrane or cytoplasmic domains, as such deletions affect the localization and expression of ORF7a. Therefore, we examined previously described naturally occurring deletions of ORF7a, Δ9nt, Δ18nt, Δ57nt, and Δ96nt, all found in the ORF7a Ig-like ectodomain and isolated from clinical samples of infected patients[50]. First, we introduced the naturally occurring ORF7a deletions in a codon-optimized ORF7a expression plasmid. We then confirmed that our ORF7a deletion variants continued to localize in cellular membranes by isolating membrane fractions followed by western blots probing for ORF7a and for glyceraldehyde-3-phosphate dehydrogenase (GAPDH) to verify the purity of our membrane fractions (Supplementary Fig. 6a). Next, we co-transfected 293T cells with SARS-CoV-2 S, pHIV-1$_{NL}$ΔEnv-NanoLuc, SERINC5 and either wild type SARS-CoV-2 ORF7a, ORF7a deletion variants (ORF7aΔ9nt, ORF7aΔ18nt, ORF7aΔ57nt, ORF7aΔ96nt) or E.V. Viruses from the culture media of the transfected cells were used to infect 293T-hACE2 cells and at 48 hpi luciferase levels were determined. We observed that all SARS-CoV-2 ORF7a naturally occurring deletion variants counteracted SERINC5 similar to wild type ORF7a (Fig. 8a).

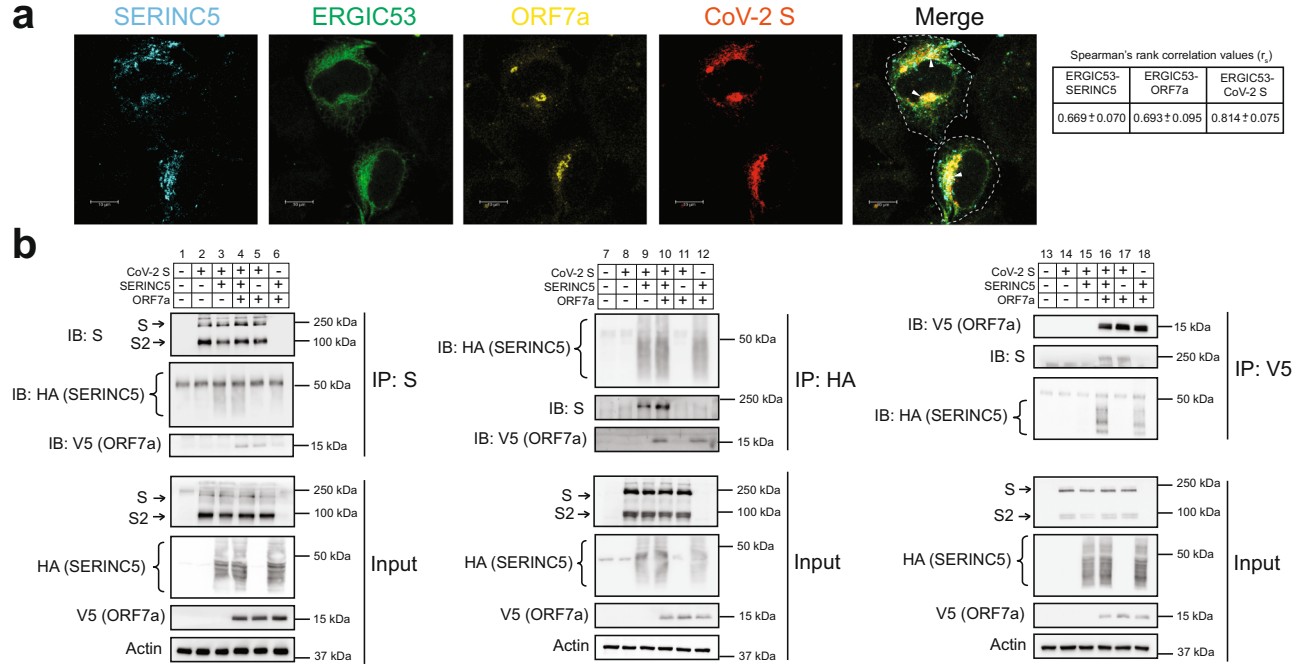

**Fig. 7 SARS-CoV-2 Spike, ORF7a and SERINC5 co-localize at the ERGIC and physically interact. a** SARS-CoV-2 Spike (S), ORF7a, SERINC5 co-localize at the ERGIC. AD-293 cells cotransfected with plasmids expressing SERINC5-HA, EGFP-ERGIC53, SARS-CoV-2 ORF7a and SARS-CoV-2 S were subjected to immunostaining. Images were acquired using 63×/1.4 oil-immersion objective with a Leica TCS SP8 confocal microscope. Arrow heads indicate areas of colocalization, and the dotted line outlines the edges of the cells. Shown are confocal images revealing co-localization of SERINC5, SARS-CoV-2 S and ORF7a with ERGIC53 cellular marker (arrows). Scale bar = 10 μm. Representative deconvolved single Z-section images are shown. Table on the right shows quantitative analyses for co-localization between SERINC5, SARS-CoV-2 S or ORF7a and ERGIC53. Spearman's rank correlation values ($r_s$, mean ± SD) of a region of interest (ROI) defined by the presence of EGFP-ERGIC53 signal were calculated using the ImageJ (FIJI) Coloc2 plugin from 3 different images each from a different experiment. **b** SARS-CoV-2 S, ORF7a, SERINC5 form a complex. 293T cells were cotransfected with SARS-CoV-2 S, SERINC5, SARS-CoV-2 ORF7a or empty vector as indicated. Cells were harvested 48 h post transfection and lysates were immunoprecipitated with anti-SARS-CoV 2 S, anti-HA and anti-V5 antibodies followed by immunoblot analyses probing with anti-SARS-CoV-2 S, anti-HA (SERINC5), anti-V5 (SARS-CoV-2 ORF7a), and anti-Actin antibodies. Representative immunoblotting results are shown. Uncropped blots are in Source Data. All results are shown for $n = 3$ independent experiments. (SARS-CoV-2 ORF7a, ORF7a; SARS-CoV-2 S, S; SARS-CoV-2 Spike S2 subunit, S2).

Thus, the naturally occurring deletions of SARS-CoV-2 ORF7a we examined do not affect its ability to counteract SERINC5.

The Ig-like ectodomain of SARS-CoV ORF7a interacts and modulates the function of a number of host factors[51]. To further understand the role of the Ig-like ectodomain of SARS-CoV-2 ORF7a in counteracting SERINC5, we generated a series of deletion mutants, in which we deleted each of the 7 β-sheets. First, we verified that the different β-sheet deletions did not alter the association of ORF7a with cellular membranes, by purifying membrane fractions from 293T cells transfected with the different β-sheet deletion ORF7a constructs, followed by western blots probing for ORF7a and GAPDH. We found that the ORF7a β-sheet deletions had no effect on the membrane association of ORF7a (Supplementary Fig. 6b). To examine the role of the ORF7a β-sheets in counteracting SERINC5, we co-transfected 293T cells with SARS-CoV-2 S, pHIV-1$_{NL}$ΔEnv-NanoLuc, SERINC5 and either wild type ORF7a, ORF7a with β-sheet deletions or E.V. Cells were harvested 48 h post transfection, viruses from the culture media of the transfected cells were used to infect 293T-hACE2 cells, and luciferase levels were measured 48 hpi. We observed that all ORF7a β-sheet deletion mutants blocked the inhibitory effect of SERINC5 similar to wild type ORF7a (Fig. 8b). Therefore, the Ig-like ectodomain is not important for the anti-SERINC5 function of ORF7a.

We also examined the role of the SARS-CoV-2 ORF7a transmembrane domain (TM) in counteracting SERINC5. To determine the importance of the ORF7a TM in its anti-SERINC5 function, we generated a chimeric SARS-CoV-2 ORF7a, in which

the transmembrane domain has been replaced with that of CD4 (ORF7aTM$^{CD4}$). We initially confirmed that ORF7aTM$^{CD4}$ still associated with cellular membranes, by purifying membrane fractions from 293T cells transfected with the ORF7aTM$^{CD4}$ followed by western blots probing for ORF7a and GAPDH. We found that the CD4 TM did not alter the membrane association of ORF7a (Supplementary Fig. 6c). Next, we co-transfected 293T cells with SARS-CoV-2 S, pHIV-1$_{NL}$ΔEnv-NanoLuc, SERINC5, and either wild type ORF7a, ORF7aTM$^{CD4}$ or E.V. Cells were harvested 48 hours post transfection, viruses from the culture media were used to infect 293T-hACE2, and luciferase levels were measured 48 hpi. We observed that ORF7aTM$^{CD4}$ did not counteract the antiviral effect of SERINC5 (Fig. 8c). Furthermore, to determine if the ORF7a TM is important for the formation the ORF7a-Spike-SERINC5 complex, we co-transfected 293T cells with SARS-CoV-2 S and SERINC5 along with either SARS-COV-2 ORF7a or ORF7aTM$^{CD4}$ and performed coIPs with anti-V5 (ORF7a). We found that ORF7aTM$^{CD4}$ no longer interacted with SARS-CoV-2 S or SERINC5 (Fig. 8d). Therefore, the ORF7a TM is critical for its anti-SERINC5 function.

**SERINC5 blocks SARS-CoV S-mediated entry and is offset by SARS-CoV ORF7a.** SARS-CoV ORF7a and SARS-CoV-2 ORF7a share 85% sequence identity[40]. As SERINC5 counteracts SARS-CoV-2 mediated entry, we speculated that due to the high sequence homology between the S and ORF7a proteins of SARS-

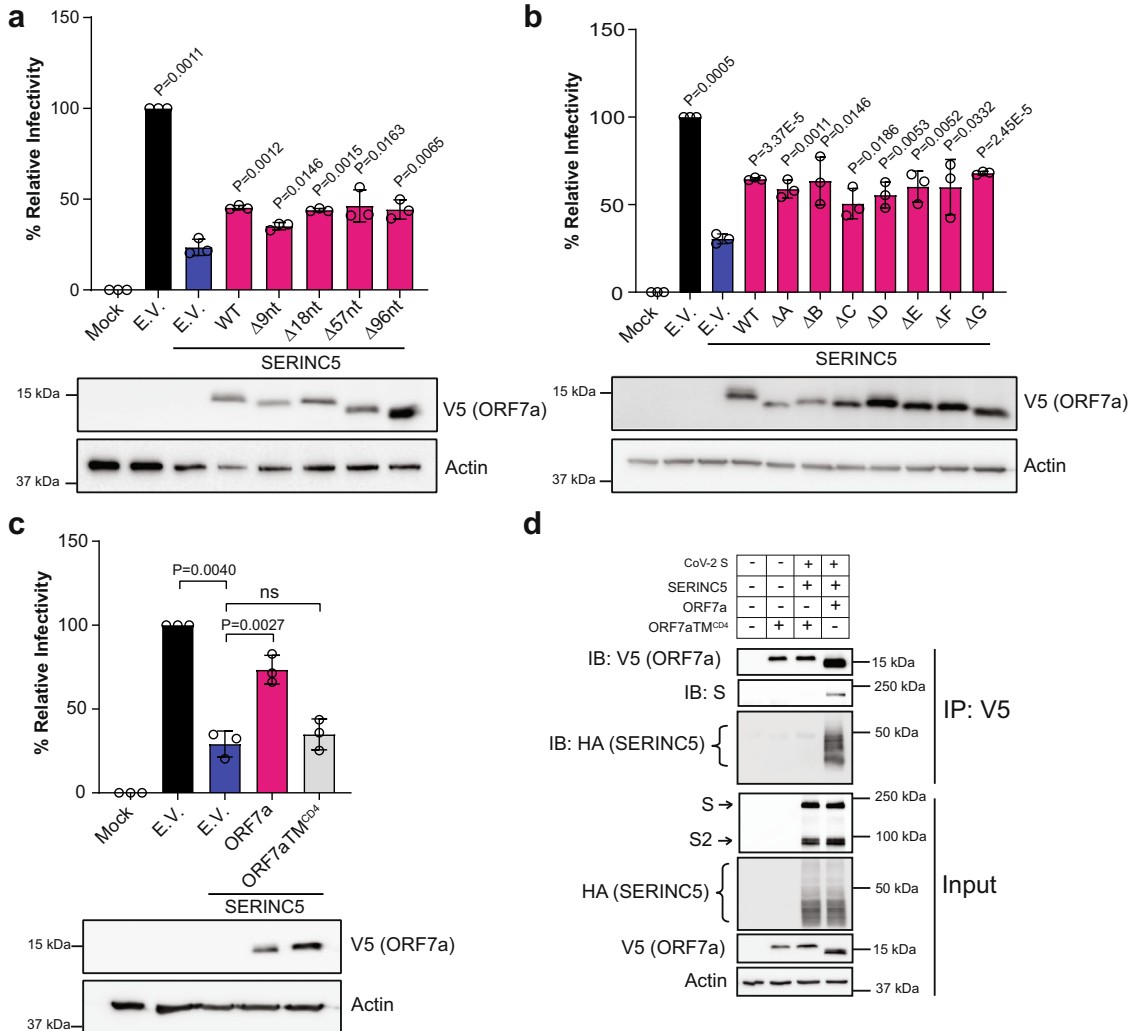

**Fig. 8 ORF7a transmembrane domain is responsible for counteracting SERINC5 restriction of SARS-CoV-2 entry.** Naturally occurring SARS-CoV-2 ORF7a deletions and the β-sheets of the SARS-CoV-2 ORF7a ectodomain do not affect its anti-SERINC5 function. 293T-hACE2 cells were infected with SARS-CoV-2 S pseudoviruses produced in the presence or absence of SERINC5 along with **a** naturally occurring deletion variants of SARS-CoV-2 ORF7a or **b** β-sheet deletion mutants of SARS-CoV-2 ORF7a. **c** The SARS-CoV-2 ORF7a transmembrane domain is critical for its anti-SERINC5 function. 293T-hACE2 cells were infected with SARS-CoV-2 S pseudoviruses produced in the presence or absence of SERINC5 along with either wild type ORF7a or ORF7-CD4 transmembrane domain chimera (ORF7aTM$^{CD4}$). For **a**, **b**, and **c**, luciferase levels were normalized to p24$^{CA}$ levels. The percentage (%) of relative infectivity with respect to pseudovirus produced in the presence of empty vector is shown. All results are presented as mean ± SD from 3 independent experiments. Statistical analysis among E.V. and SERINC5 + E.V. conditions were performed by one-sample $t$ test (two-tailed) while unpaired t-test (two-tailed) was used for comparisons among SERINC5 + E.V. and SERINC5 + ORF7a groups. Representative immunoblotting results of the expression levels of SARS-CoV-2 ORF7a variants or mutants in the producer cell lysates is shown at the bottom. **d** Transmembrane domain of SARS-CoV-2 ORF7a is required for its interaction with SARS-CoV-2 S and SERINC5. 293T cells were cotransfected with SARS-CoV-2 S, SERINC5, SARS-CoV-2 ORF7a, SARS-CoV-2 ORF7aTM $^{CD4}$or empty vector as indicated. Cells were harvested and lysates were immunoprecipitated with anti-V5 antibody followed by immunoblot analyses probing for the indicated proteins. Representative immunoblotting results from $n = 3$ independent experiments are shown. Uncropped blots are in Source Data. (empty vector, E.V.; wild type SARS-CoV-2 ORF7a, WT; SARS-CoV-2 ORF7aΔ9nt, Δ9nt; SARS-CoV-2 ORF7aΔ18nt, Δ18nt; SARS-CoV-2 ORF7aΔ57nt, Δ57nt; SARS-CoV-2 ORF7aΔ96nt, Δ96nt; SARS-CoV-2 ORF7aΔA, ΔA; SARS-CoV-2 ORF7aΔB, ΔB; SARS-CoV-2 ORF7aΔC, ΔC; SARS-CoV-2 ORF7aΔD, ΔD; SARS-CoV-2 ORF7aΔE, ΔE, SARS-CoV-2 ORF7aΔF, ΔF; SARS-CoV-2 ORF7aΔG, ΔG; SARS-CoV-2 S, S; SARS-CoV-2 Spike S2 subunit, S2; not significant, ns).

CoV and SARS-CoV-2, the antiviral function of SERINC5 may extend to SARS-CoV. To determine the role of SERINC5 on SARS-CoV entry and the anti-SERINC5 function of SARS-CoV ORF7a, we co-transfected 293T cells with SARS-CoV S and pHIV-1$_{NL}$ΔEnv-NanoLuc, along with either SERINC5, SARS-CoV ORF7a or E.V. Cells and virus from the media of the transfected cells were harvested and processed by western blots. We observed that SARS-CoV ORF7a is incorporated in purified pseudoviruses (Fig. 9a) in agreement with previous studies[41]. Furthermore, SERINC5, similar to what we saw with SARS-CoV-

2 S pseudoviruses, is also incorporated in the virions and its incorporation is not affected by the presence of SARS-CoV ORF7a (Fig. 9a). Viruses were then used to infect 293T-hACE2 cells and luciferase levels were measured 48 hpi. We found that similar to SARS-CoV-2, SERINC5 potently blocked SARS-CoV S-mediated entry (Fig. 9a) and SARS-CoV ORF7a counteracted the antiviral effect of SERINC5 (Fig. 9b). Hence, we concluded that the antiviral effect of SERINC5 is not just limited to SARS-CoV-2 but applies to other coronaviruses and that both SARS-CoV and SARS-CoV-2 ORF7a counteract SERINC5.

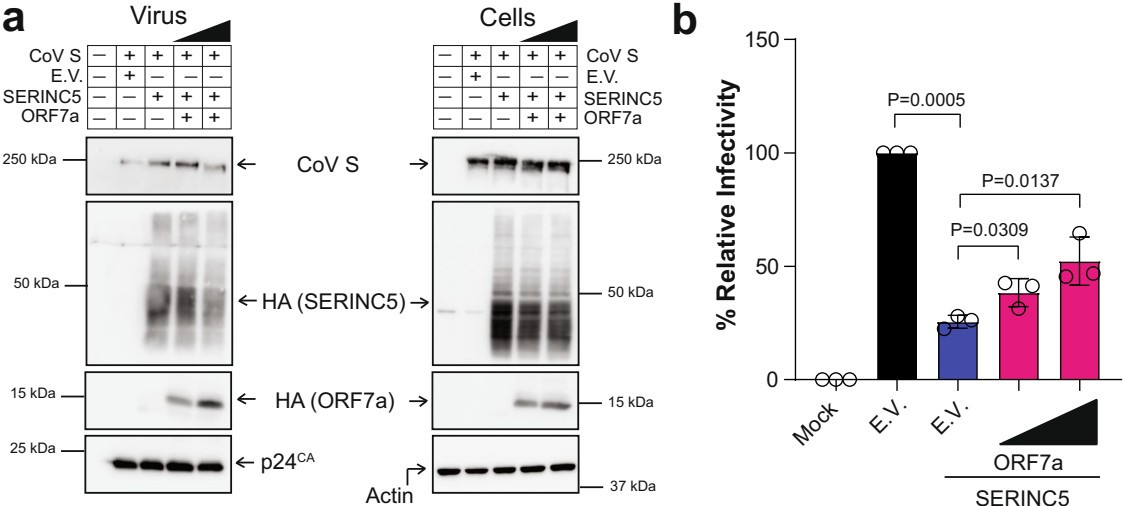

**Fig. 9 SERINC5 restricts SARS-CoV S-mediated entry and is counteracted by SARS-CoV ORF7a. a** SARS-CoV ORF7a is packaged in nascent virions and does not cause the degradation of SERINC5 or block SERINC5 incorporation in nascent virions. SARS-CoV S pseudovirus produced in the presence or absence of SERINC5 and SARS-CoV ORF7a was analyzed by immunoblotting. Representative immunoblotting results from $n = 3$ independent experiments are shown. Uncropped blots are in Source Data. **b** SERINC5 blocks SARS-CoV S pseudovirus infection and is counteracted by SARS-CoV ORF7a. 293T-hACE2 cells were infected with SARS-CoV S pseudoviruses produced in **a**. Luciferase levels were measured 48 hpi and normalized to p24$^{CA}$ levels in the input pseudovirus. The percentage (%) of relative infectivity with respect to pseudovirus produced in the presence of empty vector is shown. Results are presented as mean ± SD from $n = 3$ independent experiments. Statistical significance was determined by one-sample $t$ test (two-tailed) or unpaired $t$ test (two-tailed). (empty vector, E.V.; SARS-CoV Spike, CoV S; SARS-CoV ORF7a, ORF7a).

## Discussion

In this report, we focused on identifying a novel host restriction factor, SERINC5, of SARS-CoV-2 entry. SERINC5 is an important antiretroviral host factor both in vivo and in vitro[13,14,17] that blocks retroviral entry of a diverse number of retroviruses including HIV-1 and MLV[13,14,22].

We found that all members of the SERINC family, except SERINC4, are expressed in lung tissue and pneumocytes, natural targets of SARS-CoV-2[18]. In addition, using, SARS-CoV-2 S pseudoviruses, VLPs and infectious SARS-CoV-2, we showed that SERINC5 is incorporated inside viral particles and SERINC5 has the most robust effect on SARS-CoV-2 entry[13,14]. In our work, the appearance of SERINC5 in our western blots varies considerably, which we attribute to technical differences in the processing of the samples from experiment to experiment.

SARS-CoV-2 binds to the hACE2 receptor on the surface of the cells followed by activation of the S protein by target cell proteases resulting in viral-host membranes fusion and viral genome release inside the cell. Here we show that SERINC5 inhibits SARS-CoV-2 entry by targeting the virus-cell membrane fusion step, as is the case with retroviruses[15]. Thus, SERINC5 uses a conserved mechanism to exert its antiviral function against two disparate viral families. SARS-CoV-2 entry is dependent on the S protein. S proteins of different SARS-CoV-2 variants are quite polymorphic and show differences in entry efficiency depending on the cell line[27]. In contrast to HIV-1, for which not all HIV-1 envelopes are restricted by SERINC5[16], SERINC5 blocked SARS-CoV-2 entry for all S variants tested. Thus, our findings suggest that despite the diversity in S protein among the different SARS-CoV-2 variants, susceptibility to SERINC5 is never lost. More studies are needed to further explore the effect of SERINC5 on S proteins derived from different SARS-CoV-2 variants.

Retroviruses encode genes that counteract the deleterious effect of SERINC5. In the case of HIV-1, Nef prevents the incorporation of SERINC5 into budding virions and thus blocks the deleterious effect of SERINC5 on virus infection[13,14,22]. No viral proteins, outside of retroviruses, have hitherto been identified to counteract

SERINC5. We identified that the SARS-CoV-2 accessory protein ORF7a, a type I transmembrane protein that is highly homologous to SARS-CoV ORF7a[40], is a viral antagonist of SERINC5. SARS-CoV ORF7a has many functions in infected cells including induction of apoptosis[52,53], neutralization of Bst-2/tetherin[54] and cell cycle arrest at the G/G1 phase[55] among others. Using an array of functional and biochemical assays, we show that SARS-CoV-2 ORF7a co-localizes with and binds to SARS-CoV-2 S and SERINC5. Supporting our findings, a recent large scale proteomics analysis that suggested that SARS-CoV-2 ORF7a interacts with factors involved with lipid metabolic processes[56], which is the cellular function of SERINC5[12]. In addition, the SARS-CoV-2 ORF7a-S interaction mirrors the previously described SARS-CoV ORF7a-S interaction[41]. Nevertheless, the biophysical mechanism by which the formation of SARS-CoV-2 ORF7a/S/SERINC5 complex interferes with the antiviral function of SERINC5 remains to be elucidated. In the case of HIV-1, it is hypothesized that SERINC5 interacts with the HIV-1 envelope, resulting in conformational remodeling of the envelope and the inhibition of HIV-1 infection[57–59]. Therefore, it is possible that SERINC5 affects the structural conformation of SARS-CoV-2 S and ORF7a binds to SARS-CoV-2 S to block any such conformational changes on S. The fact that SARS-CoV-2 ORF7a, S and SERINC5 are found in both pseudoviruses as well as infectious SARS-CoV-2 virions suggests that this might be happening in the virions. Moreover, we were surprised to observe that in the context of SARS-CoV-2 infection, ORF7a blocked the incorporation of SERINC5 in budding virions, unlike what we observed with SARS-CoV-2 S pseudoviruses. This discrepancy could be due to the different sites of assembly of SARS-CoV-2 S pseudoviruses vs. infectious SARS-CoV-2 (plasma membrane vs ERGIC). In conclusion, our findings suggest that ORF7a counteracts SERINC5 in two ways; in the producer cells ORF7a prevents the incorporation of SERINC5 into SARS-CoV-2 virions and in the released virions ORF7a forms a complex with S and SERINC5 to block the antiviral effect of SERINC5 on virus-cell membrane fusion. This is in agreement with the idea that exclusion of SERINC5 from

virions is important but not adequate for alleviation of restriction of virus infection. Finally, the fact that SARS-CoV is also restricted by SERINC5 further emphasizes the importance of this host antiviral factor on coronavirus infection.

In conclusion, the antiviral effect of SERINC5 on a diverse number of viruses and the presence of SERINC5 viral antagonists demonstrate that its interplay with viral proteins is an attractive target for the development of antiviral therapies.

## Methods

**Cell culture and transfection.** 293T cells (ATCC, CRL-3216), Calu-3 cells (ATCC, HTB-55), AD-293 cells (Agilent), and HEK 293T-hACE2 (BEI Resources, NIAID, NIH, NR-52511) were cultured in Dulbecco's Modified Eagle Media (DMEM; Gibco) with 10% (vol/vol) fetal bovine serum (FBS; Sigma), and 100 mg/ml penicillin and streptomycin (P/S; Gibco). Vero E6 cells (BEI Resources, NIAID, NIH, NR-5258) were cultured in DMEM with 0.1 mM non-essential amino acids (Gibco), 1 mM sodium pyruvate (Gibco), and 100 mg/ml P/S. All transfections were performed using Lipofectamine 3000 transfection kit (Invitrogen) per manufacturer's recommendation.

**Plasmids.** MLV Gag-Pol (ΔGlyco-Gag, ΔEnv) plasmid was acquired from Alan Rein[22]. The pBJ5- SERINC1, 2, 3, 5-HA constructs were obtained from Heinrich Gottlinger[14]. The pCAGGS SARS-CoV ORF7a-HA construct was gotten from Matthew B. Frieman[54]. The pCAGGS SARS-CoV-2 S FKO-FLAG construct was obtained from Hyeryun Choe[34]. The pCG1 SARS-CoV-2 SΔ18 codon optimized variants (B.1.1.7, B.1.351, and P.1) were obtained from Stefan Pohlmann and Markus Hoffmann[27]. The pLV-SpikeV8 plasmid encoding codon-optimized Delta (B.1.67) SARS-CoV-2 SΔ19 variant was acquired from InvivoGen (p1-spike-V8). The pCAGGS SARS-CoV-2 S (Full Length), pCAGGS SARS-CoV-2 SΔ27 and pCAGGS SARS-CoV S (Full length)[60] were obtained from Paul Bates. The HIV-1 NL4-3 ΔEnv- NanoLuc, pCMV SARS-CoV-2 SΔ19 were obtained from Paul Bieniasz[21]. The pFB-*luc* construct has been previously described[61]. The following reagent was obtained through the NIH HIV Reagent Program, Division of AIDS, NIAID, NIH: HIV-1 YU2 Vpr β-lactamase expression vector (ARP-11444) contributed by Dr. Michael Miller (Merck Research Laboratories). SARS-CoV-2 E and N were PCR amplified from cDNA derived from genomic RNA of SARS-CoV-2 isolate USA-WA1/2020 (BEI Resources, NIAID, NIH, NR-52285) using primers 5′-CCCAAGCTTATGTACTCATTCGTTTCG-3′/5′-CCGCTCGAGGACCAGA AGATCAGGAACTC-3′ for E and 5′- CGCGGATCCGCCATGTCTGATAA TGGACCCCAAAATCAG-3′/5′- AAAAGCGGCCGCCGGCCTGAGTTGAGTC AGC-3′ for N followed by cloning into pCDNA3.1/myc-His A (Invitrogen) and pCDNA-V5/His TOPO (Invitrogen) respectively. Codon optimized SARS-CoV-2 ORF7a was PCR amplified from pLVX-EF1alpha-SARS-CoV-2-ORF7a-2xStrep-IRES-Puro plasmid, (Addgene, 141388, deposited by Nevan Krogan) using primers 5′-CCCAAGCTTGCCGCCACCATGAAGATCA-3′/5′-CCGCTCGAGCGCT-CAGTCTTTCTTTTCAGTG-3′ followed by cloning into pCDNA-V5/His TOPO (Invitrogen). This codon optimized SARS-CoV-2 ORF7a (WT) in pCDNA-V5/His TOPO plasmid was used as a template to generate SARS-CoV-2 ORF7a deletion variants using the Phusion SDM kit (Thermo Fischer Scientific) and primers listed in Supplementary Table 1, and to obtain a SARS-CoV-2 ORF7a expression plasmid with a V5 tag at the N′ terminus, in which the V5 tag was placed immediately downstream of the signal peptide sequence followed by the removal of the C-terminal V5/His tag using the Phusion SDM kit and primers listed in Supplementary Table 1. SARS-CoV-2 ORF7aTM[CD4] was created by replacing the transmembrane domain of ORF7a with that of CD4 using the NEBuilder HiFi DNA assembly kit (New England Biolabs) with the following primers (CD4TM_7a-F: 5′-GAGGTGCAAGAGATGGCCCTGATTGTGCTG-3′/CD4TM_7a-R: 5′-AGTCTTTCTTTTGAAGAAGATGCCTAGCCCAATG-3′, 7a_delTM-F: 5′-GGCATCTTCTTCAAAAGAAAGACTGAGCGCTC-3′/7a_delTM-R: 5′-AATCAGGGCCATCTCTTGCACCTCCTCTTG-3′). pLVX-EF1a-EGFP-ERGIC53-IRES-Puromycin was acquired from Addgene (134859, deposited by David Andrews). pCMV6-TMED2-FLAG was acquired from Ori-Gene (RC206849).

**Immunoblotting.** Cells lysates were prepared as previously described[17,62] followed by western blots using the following antibodies: mouse anti-SARS-CoV/SARS-CoV-2 S A-19 (1: 5000, GeneTex), mouse anti-SARS CoV/SARS-CoV-2 ORF7a 3C9 (1: 1000, GeneTex), mouse anti-V5 (1: 5,000, Thermo Fisher Scientific), rat anti-MLV p30 R187 (1:1000, ATCC, CRL-1912), rabbit anti-HA C29F4 (1: 2,000, Cell Signaling Technology), mouse anti-HA 2-2.14 (1: 5000, Invitrogen), rabbit anti-FLAG D6W5B (1: 2,000, Cell Signaling Technology), rabbit anti-GAPDH 14C10 (1: 3000, Cell Signaling Technology), rabbit anti-SARS-related Coronavirus Nucleocapsid protein 019 (1: 2,000, BEI Resources, NIH, NIAID, NR-53793), mouse anti-HIV-1 p24 AG3.0 (1: 2000, NIH/AIDS Reagent Program, ARP-4121), monoclonal anti-β-actin (1: 7,000, Sigma-Aldrich). Horseradish peroxidase (HRP)-conjugated anti-rabbit IgG (1: 2,000, Cell Signaling Technology), HRP-conjugated anti-rat IgG (1: 2000, Cell Signaling Technology) and HRP-conjugated anti-mouse IgG (1: 7,000, EMD Millipore) were used for detection using the enhanced

chemiluminescence detection kits Clarity and Clarity Max ECL (Bio-Rad). Quantitation of bands in western blots were performed using the ImageJ software (National Institutes of Health; https://imagej.nih.gov/ij/).

**Expression analysis of *SERINC* genes.** cDNA was synthesized using the Super-Script III First Strand Synthesis kit (Invitrogen) per manufacturer's recommendation using 2 μg of the total lung RNA purified from 3 human donors (OriGene Technologies; CR560789, CR561388 and CR562269). RT-qPCR was performed using the Power Up SYBR Green PCR master mix (Applied Biosystems) in a CFX384 Touch Real-Time PCR detection system (Bio-Rad). Primers used were: *SERINC1* 5′-AGATAATGAAAGGGATGGTGTC-3′/5′-ACAGCACGATGCCA ATCCAACT −3′, *SERINC2* 5′- TGGTGCTGCTCATCGACTTT-3′/5′-TGAAG AAGAAGAGCCTGCG-3′, *SERINC3* 5′- AATTCAGGAACACCAGCCTC-3′/5′-GGTTGGGATTGCAGGAACGA-3′, *SERINC4* 5′- CGTCCTCCAGAGAGAGT AATCC-3′/5′- CCACAGGGGTCCAAATACCTC-3′, *SERINC5* 5′- ATCGAG TTCTGACGGCTCTGC-3′/5′-GCTCTTCAGTGTCCTCTCCAC-3′, and *GAPDH* 5′-AACGGGAAGCTTGTCATCAATGGAAA-3′/5′-GCATCAGCAGAGGG GGCAGAG-3′ for normalization.

Calu-3 cells ($5 \times 10^4$/well) were seeded in a 96-well plate. The following day, RNA was isolated using RNeasy Mini kit (Qiagen) followed by cDNA synthesis and RT-qPCR as described above.

**SARS-CoV-2 infection and interferon treatment of Calu-3 cells.** Calu-3 cells were seeded at a density of $0.2 \times 10^6$/well in a 24-well plate. Forty-eight hours later, cells were either mock infected or infected with SARS CoV-2 virus (isolate USA-WA1/2020; see *SARS-CoV-2 Infectious virus production* section) at 5 MOI and harvested 2 and 6 hpi. Infection experiments were performed in a biosafety level 3 laboratory. RNA isolation, cDNA synthesis and RT-qPCR were performed as described above (see *Expression analysis of SERINC genes* section). In addition to *SERINC1, 2, 3, 5* and *GAPDH* primers, the following previously described primers[63] were used for measuring SARS CoV-2 genomic RNA (gRNA) CoV2U TR-FP: 5′- AAAATCTGTGTGGCTGTCACT-3′ and CoV2UTR-RP: 5′- GACG AAACCGTAAGCAGCCT-3′.

Calu-3 cells ($5 \times 10^4$/well) were seeded in a 96-well plate and treated with or without 500 U/ml of human Interferon Beta 1b (PBL Assay Science). Cells were harvested at 4, 8, 16, 24 h post treatment. RNA isolation, cDNA synthesis and RT-qPCR were performed as mentioned above. Primers for *SERINC1, 2, 3, 5,* and *GAPDH* are described above, *Interferon-stimulated gene 15* (*ISG15*) primers are 5′- GATCACCCAGAAGATCGGCG-3′/5′- GGATGCTCAGAGGTTCGTCG-3′.

**Pseudovirus production.** 293T cells were seeded in a six-well plate ($0.5 \times 10^6$ cells/well) a day prior to infection. To generate SARS-CoV-2 S pseudotyped (MLV/Luc) virions, cells were co-transfected with plasmids for MLV Gag-Pol (ΔGlyco-Gag, ΔEnv) (2 μg), pFB Luc (2 μg), pCAGGS SARS-CoV-2 SΔ27 (10 ng) and either pBJ5-SERINC3, 5-HA (0.5 μg) or empty vector using Lipofectamine 3000 (Thermo Fisher Scientific) per manufacturer's recommendation. To generate SARS-CoV-2 S pseudotyped (HIV/NanoLuc) virions, cells were co-transfected with plasmids for HIV-1[NL4-3] ΔEnv- NanoLuc (2.5 μg), pCMV SARS-CoV-2 SΔ19 (0.73 μg) and either pBJ5-SERINC1, 2, 3, 5-HA (0.5 μg) or empty vector using Lipofectamine 3000 (Thermo Fisher Scientific) per manufacturer's recommendation. For all HIV/NanoLuc SARS-CoV-2 S pseudoviruses we used the pCMV SARS-CoV-2 SΔ19 plasmid unless stated otherwise.

Pseudoviruses (HIV/NanoLuc) with SARS-CoV-2 S from different variants were generated as above using pCG1 SARS-CoV-2 SΔ18 variants (B.1.1.7, B.1.351 and P.1; 0.3 μg each) or pLV-SpikeV8 SARS-CoV-2 SΔ19 variant (B.1.617; 0.73 μg). Pseudoviruses (HIV/NanoLuc) with SARS-CoV-2 S FKO were produced as above using pCAGGS SARS-CoV-2 S FKO (0.65 μg). SARS-CoV S pseudoviruses (HIV/NanoLuc) were produced as described above using pCAGGS SARS-S Full length (0.7 μg) with or without pBJ5-SERINC5-HA (0.5 μg). For all pseudovirus production experiments, culture media was removed and replenished 24 h after transfection. Cells and culture supernatants were harvested 48 h post transfection. Cells were lysed and processed for immunoblotting (see *Immunoblotting* section). Culture supernatants were cleared of cell debris through centrifugation at $714 \times g$ for 10 min at 4 °C followed by filtration through 0.45 μm filter. Aliquots of filtered culture supernatants were either stored at −80 °C for future experiments or pelleted through a 30% sucrose cushion as previously described followed by storage at −80 °C[64].

**SARS-CoV-2 infectious virus production.** The following reagents were obtained through BEI Resources, NIAID, NIH: SARS-Related Coronavirus 2, Isolate USA-WA1/2020; NR-52281[20], SARS-Related Coronavirus 2, Isolate USA-WA1/2020, Recombinant Infectious Clone (icSARS-CoV-2-WT);[42] NR-54001 and SARS-Related Coronavirus 2, Isolate USA-WA1/2020 ΔORF7a, Recombinant Infectious Clone with Enhanced Green Fluorescent Protein (icSARS-CoV-2-eGFP/ΔORF7a);[42] NR-542002. Infection experiments using SARS-CoV-2 infectious viruses were performed in a biosafety level 3 laboratory at the University at Buffalo, Jacobs School of Medicine and Biomedical Sciences, Buffalo, NY. Viruses were propagated on Vero E6 cells and culture supernatants were harvested at 48 hpi. All experiments were performed using this early passage (p1) viruses. For calculation

of viral RNA copies, 100 μl of culture supernatant was used for RNA isolation followed by cDNA synthesis and RT-qPCR. The following primers were used for SARS-CoV-2 S detection: 5′- CCTACTAAATTAAATGATCTCTGCTTTACT-3′/ 5′-CAAGCTATAACGCAGCCTGTA-3′.

SARS-CoV-2 infectious viruses containing SERINC3 or SERINC5 were generated by infecting 293T-hACE2 cells transfected with SERINC3 or SERINC5 with SARS-CoV-2 virus (isolate USA-WA1/2020). Cells and culture supernatants were harvested 48 hpi. Cells were lysed in 1× RIPA buffer and processed for immunoblotting (see *Immunoblotting* section). Culture supernatants were cleared of cell debris and aliquots were either stored at −80 °C for future experiments or pelleted through a 30% sucrose cushion followed by lysis in 1× RIPA buffer for immunobotting.

For infectivity experiments with SARS-CoV-2 infectious viruses containing SERINC3 or SERINC5, culture supernatants normalized for viral RNA copies were used to infect Calu-3, 293T-hACE2 or *Mus dunni* cells ($5 \times 10^4$ cells/well). After 1 h adsorption at 37 °C, cells were washed 3× with PBS and maintained in culture media containing 2% FBS. At 6 hpi, cells were washed again 3× with PBS followed by RNA isolation, cDNA synthesis and RT-PCR as mentioned above (see *SARS-CoV-2 infection and interferon treatment of Calu-3 cells* section). The following primers were used for detection of mouse *GAPDH*: 5′-CCCCTTCATTGACCT CAACTACA-3′/5′-CGCTCCTGGAGGATGGTGAT-3′.

**Pseudovirus entry assays**. For infections using the SARS-CoV-2 S MLV pseudotypes, 293T-hACE2 cells ($2.5 \times 10^4$ cells/well) were seeded in a 24-well plate. Cells were infected the following day and lysed 48 hpi followed by measuring luminescence using the Steady-Glo luciferase assay system (Promega) per manufacturer's recommendation and Biostack4 (BioTek) luminometer, an automated plate reader. For SARS-CoV-2 S/CoV S HIV pseudotypes, Calu-3 or 293T-hACE2 cells ($2.5 \times 10^4$ cells/well) were seeded in a 96-well plate. Cells were infected the next day and lysed 48 hpi followed by measuring luminescence using the Nano-Glo luciferase system (Promega) per manufacturer's recommendation and a Biostack4 (BioTek) luminometer. Infectivity was determined by normalizing the luciferase signals to virus levels as determined by western blots probing for MLV p30$^{CA}$ or HIV p24$^{CA}$ on culture supernatants (see *Immunoblotting* section).

**VLP production**. SARS-CoV-2 VLPs were produced as previously described[26]. Briefly, 293T cells ($0.5 \times 10^6$ cells/well) were seeded in 6-well plate and co-transfected with plasmids encoding SARS-CoV-2 SΔ19 (1.1 μg), SARS-CoV-2 M (0.7 μg), SARS-CoV-2 N (0.7 μg), SARS-CoV-2 E (0.72 μg), and either SERINC5-HA (0.5 μg) or empty vector (0.5 μg). Culture media was replaced 24 h post transfection. Cells and culture supernatants were harvested 48 hours post transfection. Cells were lysed and processed for immunoblotting (see *Immunoblotting* section). Following ultracentrifugation, VLPs were stored at −80 °C for future use.

**Binding assay with FITC-labeled SARS-CoV-2 S pseudovirus**. 293T-hACE2 or 293T cells ($0.2 \times 10^6$ cells/well) were seeded in a 24-well plate. Concentrated SARS-CoV-2 S HIV pseudotyped viruses generated as mentioned above (see *Pseudovirus production* section) were labeled with FITC using EzLabel protein FITC Labeling Kit (BioVision) per manufacturer's recommendation. After p24$^{CA}$ normalization, labeled virions were added to cells and incubated on ice for 1 h. Cells were shifted to 1 h at 37 °C, washed 3× in ice cold 1× PBS, lifted using 100 μl of Versene (Gibco), fixed with 4% paraformaldehyde for 10 min at 4 °C, and acquired on BD LSRFortessa using BD FACSDiva 8.0.2 followed by analysis using FlowJo version 10.8.0.

**Cathepsin L treatment experiment**. SARS-CoV-2 S FKO-FLAG pseudotyped HIV viruses were generated in the presence or absence of SERINC5-HA as mentioned above (see *Pseudovirus production* section). As previously described[35], virus pellets were resuspended in 100 μl of 1× PBS with Ca$^{2+}$ and Mg$^{2+}$, pH 5.5 and either treated with PBS (mock) or human Cathepsin L (10 μg/ml, Millipore Sigma, 219402) for 1 h at room temperature (RT). We used 20 μM of Cathepsin L inhibitor SID 26681509 (MedChemExpress) as a control. Virus supernatants were then resolved on 8% SDS-PAGE gels. SARS-CoV-2 S FKO was detected using rabbit anti- FLAG as described above (see *Immunoblotting* section).

**BlaM-Vpr fusion assay**. To generate BlaM-Vpr containing SARS-CoV-2 S HIV pseudotyped viruses in the presence or absence of SERINC5, transfections were performed as described above (see *Pseudovirus production* section) along with 2.5 μg of HIV-1 YU2 Vpr β-lactamase expression plasmid. Culture supernatants were harvested 24, 48, and 72 h post transfection, pooled, and pelleted by ultra-centrifugation at $133,900 \times g$ for 2.5 h at 4 °C. Pseudoviruses were resuspended and stored at −80 °C. Virion fusion assays were performed as previously described[36,37] with some modifications. Briefly, 293T-hACE2 cells ($5 \times 10^4$ cells/well) were seeded in a 96-well plate a day prior to infection. Cells were infected with 150 ng of p24$^{CA}$ equivalent virions per well, centrifuged at $257 \times g$ for 2 h at 22 °C and further incubated at 37 °C for 1 h. When using a neutralizing antibody, cells were infected with virions pre-incubated for 1 h at 37 °C in the presence of a monoclonal anti-SARS-CoV-2 Spike glycoprotein receptor binding domain, chimeric potent neutralizing antibody (1 μM) (BEI Resources, NIH, NIAID, NR-55410). Cells were

washed in 200 μl of CO$_2$-independent media (Gibco) by centrifuging at $365 \times g$ for 5 min at RT and were resuspended in 100 μl of CO$_2$-independent media followed by staining with 20 μl of 6× CCF2-AM substrate loading solution (Invitrogen, K1032) for 1 h at RT in the dark. Cells were then washed twice in 250 μl of development media [CO$_2$-independent media containing 2.5 mM probenecid (Sigma)] by centrifuging at $365 \times g$ for 5 min at RT, resuspended in 400 μl of development media and incubated at RT for 16 h in the dark. Subsequently, cells were washed, fixed and the change in emission fluorescence of CCF2-AM after cleavage by the BlaM-Vpr post-virion fusion was monitored via BD LSRFortessa using BD FACSDiva 8.0.2 followed by data analysis using FlowJo version 10.8.0.

**Effect of SARS-CoV-2 ORF7a on SARS-CoV-2 entry**. To determine the effect of SARS-CoV-2 ORF7a on SERINC5-mediated restriction, SARS-CoV-2 S pseudo-typed virions (MLV/Luc and HIV/NanoLuc) were produced in 293T cells in the presence of pBJ5-SERINC5-HA (0.5 μg) and pCDNA SARS-CoV-2 ORF7a V5/His (1.0 μg for MLV/Luc and 0.25–0.75 μg for HIV/NanoLuc) as described above. For virion incorporation of SARS-CoV-2 ORF7a, 293T cells seeded in 10 cm culture dishes were co-transfected as described above scaling up by 7-fold the amount of DNA. For the different deletion variants of SARS-CoV-2 ORF7a we generated SARS-CoV-2 S pseudoviruses (HIV/NanoLuc) in the presence of pBJ5-SERINC5-HA as described above using the following amounts of SARS-CoV-2 ORF7a: 3.5 μg of ORF7aΔ9nt, 2.5 μg of ORF7aΔ18nt, 3.0 μg of ORF7aΔ57nt, 3.0 μg of ORF7aΔ96nt, 3.5 μg of ORF7aΔA, 3.5 μg of ORF7aΔB, 2.0 μg of ORF7aΔC, 2.0 μg of ORF7aΔD, 3.5 μg of ORF7aΔE, 3.0 μg of ORF7aΔF, and 3.5 μg of ORF7aΔG. In the case of SARS-CoV ORF7a, we generated SARS-CoV S pseudoviruses as described above with either pCAGGS SARS-CoV ORF7a-HA (5–8 ng) or empty vector.

**SARS-CoV-2 cell spread assay**. 293T-hACE2 cells ($0.2 \times 10^6$ cells/well) in a 12-well plate were transfected with either pBJ5-SERINC5-HA (1.5 μg) or empty vector. Next day, cells were infected with SARS-CoV-2 WT or SARS-CoV-2 eGFP (ΔORF7a) virus (0.1 viral RNA copy per cell). Cells were harvested 24, 36, and 48 hpi followed by RNA isolation, cDNA synthesis and RT-qPCR as mentioned above (see *SARS-CoV-2 infection and interferon treatment of Calu-3 cells* section).

**SARS-CoV-2 ORF7a trans-complementation assay**. 293T-hACE2 cells ($0.5 \times 10^6$ cells/well) in a 6-well plate were transfected with either pBJ5-SERINC5-HA (1.5 μg) or empty vector along with pCDNA SARS-CoV-2 ORF7a (2.5 μg). Next day, cells were infected with SARS-CoV-2 eGFP (ΔORF7a) virus. Cells and culture supernatants were harvested 48 hpi and processed as described above (see *SARS-CoV-2 infectious virus production* section).

For infectivity experiments, culture supernatants normalized for viral RNA copies were used to infect 293T-hACE2 or *Mus dunni* cells ($1 \times 10^5$ cells/well). RNA was isolated 6 hpi, from the infected cells and processed as mentioned above (see *SARS-CoV-2 infection and interferon treatment of Calu-3 cells & SARS-CoV-2 infectious virus production* sections).

**siRNA knockdown of SERINC5 in Calu-3 cells followed by SARS-CoV-2 infection**. Calu-3 cells ($1 \times 10^5$ cells/well) were seeded in a 96-well plate and reverse-transfected with 6 pmol of either negative control siRNA (Thermo Fisher Scientific, AM4611) or SERINC5-specific siRNA (Thermo Fisher Scientific, s48829, Sense: 5′-GGGUCAUUUAUGACGAGAATT-3′/antisense: 5′-UUCUCGUCAU AAAUGACCCGT-3′). Forty-eight hours post-transfection, cells were infected with SARS-CoV-2 WT or SARS-CoV-2 eGFP (ΔORF7a). At 24 hpi, culture super-natants were harvested and used for virion infectivity assays as mentioned above (see *SARS-CoV-2 infectious virus production* section). siRNA knockdown ver-ification of the siRNA transfected Calu-3 cells was performed 72 h post transfec-tion, by harvesting RNA, followed by cDNA synthesis and RT-qPCR as described above (see *Expression analysis of SERINC genes* section).

**shRNA knockdown of SERINC5 in Calu-3 cells followed by SARS-CoV-2 infection**. The following oligos were used to construct pLKO.1 (Addgene) expressing shRNA to SERINC5 (shS5): S5sh4F 5′- CCGGAAGATCGAGTTCT GACGCTCTCTCGAGAGAGCGTCAGAACTCGATCTTTTTTTG-3′/S5sh4R 5′- AATTCAAAAAAAGATCGAGTTCTGACGCTCTCTCGAGAGAGCGTCAG AACTCGATCTT-3′. Lentiviruses were generated using shS5 or negative control vector containing scramble shRNA (shCon; Addgene) as per the manufacturer's instruction and used to transduce Calu-3 cells at a MOI of 0.01. After lentiviral transduction, cells were maintained in media supplemented with 4 μg/ml pur-omycin (Research Products International). Knockdown verification of puromycin selected Calu-3 cells was performed as described above (see *Expression analysis of SERINC genes* section).

Calu-3 cells stably expressing shS5 or shCon ($0.5 \times 10^5$ cells/well) in a 96-well plate were infected with SARS-CoV-2 WT or SARS-CoV-2 eGFP (ΔORF7a) virus (0.02 viral RNA copy per cell). Cells were harvested 24 and 48 hpi followed by RNA isolation, cDNA synthesis and RT-qPCR as mentioned above (see *SARS-CoV-2 infection and interferon treatment of Calu-3 cells* and *SARS-CoV-2 infectious virus production* sections).

**Immunofluorescence**. 12-mm coverslips (Carolina, 633029) were treated with 0.01% poly-L-lysine solution (Sigma Aldrich) for 30 min at RT, dried and seeded with AD-293 cells ($5 \times 10^4$ cells cells/well). The following day, cells were co-transfected with pLVX-EF1a-EGFP-ERGIC53-IRES-Puromycin (250 ng), pCMV SARS-CoV-2 SΔ19 (250 ng), pCDNA-SARS CoV-2 ORF7a-V5 (500 ng) and pBJ5-SERINC5-HA (125 ng) plasmids. At 24 h post transfection, cells were washed, fixed with 4% paraformaldehyde and permeabilized with 0.3% Triton X-100 (Fischer Scientific) for 5 min at RT. Cells were then blocked with 1× PBS containing 4% bovine serum albumin (Research Products International) and 0.075% Tween 20 (Research Products International) for 1 h at RT. Cells were incubated with mouse anti-V5 (ORF7a detection) (1:300- Thermo Fisher Scientific, R960-25), rabbit anti-HA (SERINC5 detection) (1: 250- Signaling Technology, 3724) and monoclonal anti-SARS Coronavirus Recombinant Human IgG1, clone CR3022 (1: 250- BEI Resources, NIAID, NIH, NR-53876) in blocking buffer overnight at 4 °C. Cells were then stained with Alexa Fluor 350 goat anti-Rabbit IgG, (1: 2000- Invitrogen, A-11046), Alexa Fluor 594 goat anti-mouse IgG (1: 2000- Invitrogen, A-11005) and Alexa Fluor 633 goat anti-human IgG (1: 1500- Invitrogen, A-21091) diluted in blocking buffer for 1 h at RT. Subsequently, cells were washed 3× in 1× PBS and mounted in antifade mounting media (0.25% 1,4-Phenylenediamine and 90% Glycerol in 1× PBS) prior to imaging. Images were acquired using 63×/1.4 oil-immersion objective with a Leica TCS SP8 confocal microscope (Leica; Buffalo Grove, IL). Multiple random fields with transfected cells were selected for each condition from three independently performed experiments. For each field, a Z-series of images was acquired, deconvolved using Leica Lightning deconvolution software and brightness was adjusted using Leica LAS X software 3.7.4.23463. Quantitative analyses for co-localization between SERINC5, SARS-CoV-2 S or ORF7a and ERGIC53 were performed using a region of interest (ROI) defined by the presence of EGFP-ERGIC53 signal using the ImageJ (FIJI) Coloc2 plugin from three different images each from a different experiment.

**CoIPs**. 293T cells were seeded in a 10 cm culture dish ($3.5 \times 10^6$ cells) and co-transfected with pCMV SARS-CoV-2 SΔ19 plasmid (5.1 μg) and either pBJ5-hSERINC5-HA (3.5 μg), pCDNA SARS-CoV-2 ORF7a-V5 or N'V5-ORF7a (7.0 μg). Cells were washed and lysed in 1 ml of NP-40 lysis buffer (Research Products International) containing 1× Halt Protease and phosphatase inhibitors (Thermo Fischer Scientific). Following centrifugation, clarified lysates were used for immunoprecipitation. CoIPs were performed using the Dynabeads protein A immunoprecipitation kit (Thermo Fisher Scientific) per manufacturer's instructions with modifications. Briefly, 50 μl Dynabeads were preincubated with either mouse anti-V5 (1:400) (Thermo Fisher Scientific, R960-25), rabbit anti-HA C29F4 (1:50) (Cell Signaling Technology, 3724) or rabbit anti-SARS-related Coronavirus 2S Glycoprotein (1:100) (BEI Resources, NIAID, NIH, NR-53788) antibodies. Cell lysates (1000 μg) were incubated with antibody coated Dynabeads for 1 h at RT followed by overnight incubation at 4 °C. For coIPs with pCDNA SARS-CoV-2 N'V5-ORF7a transfections, cell lysates (2000 μg) were incubated with antibody coated Dynabeads for 1 h at RT. Dynabeads were then washed and eluted. The eluted fractions were subjected to western blot analysis. Antibodies used to probe western blots are described above (see *Immunoblotting* section).

For coIPs including SARS-CoV-2 ORF7aTM[CD4] and TMED2, 293T cells seeded in a six-well culture dish ($0.5 \times 10^6$ cells) and co-transfected with SARS-CoV-2 ORF7aTM[CD4] (1.0 μg) or N'V5-ORF7a (1.0 μg) and either pBJ5-hSERINC5-HA (0.5 μg), pCMV SARS-CoV-2 SΔ19 plasmid (0.73 μg) or pCMV6-TMED2-FLAG (5 ng). Cell lysates (500 μg) were incubated with antibody coated Dynabeads for 1 h at RT followed by western blot analysis as described above.

**Membrane fractionation assay**. To verify that SARS-CoV-2 ORF7a mutants and the SARS-CoV-2 ORF7aTM[CD4] chimera localized to the plasma membrane, we used the Mem-PER Plus membrane extraction kit (Thermo Fisher Scientific) per manufacturer's recommendation. Membrane fraction purity was verified by western blots probing for rabbit anti-GAPDH (see *Immunoblotting* section).

**Statistical analysis**. Statistical analyses were performed using GraphPad Prism software version 9.0.2. Statistical tests used to determine significance are described in the figure legends. A difference was considered to be significant if the $p$ value was <0.05.

**Reporting summary**. Further information on research design is available in the Nature Research Reporting Summary linked to this article.

## Data availability
The data supporting the findings of this study are available from the corresponding author upon request. Source data are provided with this paper.

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

## Acknowledgements

We thank Paul Bates, Amy Jacobs, Paul Bieniasz, Matthew Frieman, Heinrich Gottlinger, Hyeryun Choe, Markus Hoffmann, Stefan Pohlmann, and Carlos Baptista for kindly and generously providing us with reagents for this project. Melanie Goolsby and Christelle Roux, who assisted us with our initial infection experiments. We also thank David Paulowski for BSL-3 training and Wade Sigurdson at the Confocal Microscopy and Flow Cytometry Core facility of the Jacobs School of Medicine for his technical assistance. This work was supported by National Institutes of Health Grant R21 AI144147-A1 to S.S., and R56 AI165161 to S.S.

## Author contributions

U.T. and S.S. contributed to study design and literature research. U.T., S.U., E.B.I and B.W. were involved in the experimental studies. U.T., S.U., E.B.I and S.S. performed data analysis/interpretation. U.T. and S.U. performed statistical analysis: U.T. and S.S were involved in manuscript preparation—original draft and manuscript editing.

## Competing interests

The authors declare no competing interests.
