## [Peer Review File · Nature Communications]

Reviewer comments, first round review –

Reviewer #1 (Remarks to the Author):

In the manuscript by Timilsina et al., the authors show that the protein SERINC5 is able to restrict SARS-CoV-2 infection in much the same way as it has been shown to restrict HIV-1 infection. They find this to be the case when using a SARS-CoV-2 Spike pseudotyped lentiviral reporter system, as well as the authentic SARS-CoV-2 virions. Furthermore, the authors attempt to illustrate a possible mechanism whereby SERINC5 might antagonise viral infection, while providing convincing data that the SARS-CoV-2 ORF7a counteract the action of SERINC5. The latter is perhaps the highlight of the paper.

Overall, Timilsina et al. provide a thorough and well-designed study that sheds light on an as-yet uncharacterised virus-host interaction system by using a range of careful assays. However, in some cases the authors overstate their conclusions in a way that is not sufficiently supported by the data presented.

Major concerns

1. The authors do not clearly explain whether their statistical testing is performed on the raw or normalised values, such as in Fig. 1B, E and Fig. 3 and Fig. 5A, E where a specific treatment has been set to 100% and all other measurements are in relation to this. The authors indicate that they have used a parametric t-test – however, if the testing has been performed on the normalised data, this indicates a violation of the assumption of parametric testing (equal variance between groups) because normalisation in this way would have set the variation of the reference group to 0.
2. The authors should make it clear that the detection of SERINC5 in western blots is made possible by expressing a SERINC5-FLAG fusion protein, which is then detected by an anti-FLAG antibody. In Fig. 2A and C, for example, this is not indicated but rather mentioned later in the Methods section. It is important to specify that the detected SERINC5 is not the authentic protein but a recombinant version.
3. Where the authors have used the authentic SARS-CoV-2 virus, they have indicated infection levels as relative levels of S mRNA to GAPDH mRNA, such as Fig. 1C. This is an unusual way to represent SARS-CoV-2 infection. Instead, a more convincing and rigorous approach would have been to use a flow cytometric assay to assess how many cells are positive for the SARS-CoV-2 N protein, RT-qPCR for relative copies of genomic RNA or plaque assays using the virions released into the supernatant. As it is, the data makes it difficult to estimate the level of viral infection in the system. A lack of change in SERINC5 levels, for example, could in fact be due to very low levels of infection.
4. The authors should quantify their western blot data before claiming to see a change in the level of a protein. For example, in Fig. 5 E, the authors claim that the level of SERINC5 is reduced in the virions, yet they show only one representative western blot without replicates or quantitative data. Claims like these should be backed up with the appropriate replicated measurements and statistical testing.
5. Throughout the paper, the appearance of SERINC5 in the western blots displayed varies considerably. In Fig. 2A, it appears as two bands, in Fig. 5C as multiple different bands and Fig. 6B as a smear. The authors should address this in the text, at the very least, as this differing appearance makes interpretation of the results difficult.
6. The authors claim that the antiviral action of SERINC5 is via antagonisation of S mediated fusion (line 246-247). Firstly, this assertion is too definitive given the data at hand. What would be needed is, for example, the same experiment (Fig. 4C) in the presence of a neutralising Spike antibody to illustrate that what is being inhibited is specifically S function. Furthermore, the authors should explain more clearly in the text as to why they decided to investigate cathepsin/TMPRSS2-mediated S cleavage and not furin-mediated cleavage (Fig 4B).
7. In Fig. 6A, the authors should provide a brightfield version of the microscopic images. Showing only the fluorescent channels makes it difficult to understand exactly what is being depicted, i.e. how many cells are in the image and where the cell membranes are located. Furthermore, the co-localisation of the proteins in the images should be quantified using the appropriate software to support the qualitative data further.

Minor concerns

1. The authors refer vaguely to “databases of peripheral blood mononuclear cells (PBMCs)” (lines 101-102) but fail to explain what databases these are (transcriptomic?). Furthermore, the citation provided (numbered 19) does not make clearer what databases are being referenced. The authors should be more explicit in the text about this.
2. The authors should consider revising the grammar and language usage throughout the paper. For example, “we concluded that 159 SERINC5 gets incorporated in SARS-CoV-2 VLPs” (lines 158-159) is fairly informal, unacademic phrasing and “gets” should be replaced with “is”. The authors could reconsider the usage of the verb “get” throughout the paper.
3. SERINC5 is designated an “attractive target for the development of antiviral therapies” (line 461). This phrasing is confusing as antiviral therapies are targeted against a virus and not a host factor. Rephrasing and revision is advised.
4. It should be made more clear in the main body of the text which virus strain has been used for which assay. This is discussed in the methods section (lines 576-601) but, worryingly, the authors fail to cite any papers or works outlining the origin of the different virus strains. Furthermore, it is generally advised to sequence the genomes of viral isolates via NGS to verify that the virus in use does not harbour any mutations that would indicate considerable variation from the original isolate. The furin cleavage site, for example, is often mutated in SARS-CoV-2 virions that have been propagated in cell culture. The authors do not provide this.

Reviewer #2 (Remarks to the Author):

HIV-1 Nef is known to enhance the infectivity of progeny virions by inhibiting the uptake of the host multipass transmembrane protein SERINC5 into virions. In the present study, the authors first show that several members of the SERINC family, including SERINC5, are expressed in a human lung cancer cell line. They then go on to show that SERINC5 can inhibit SARS-CoV-2 Spike-mediated entry by targeting virus-cell membrane fusion, both in the context of HIV-based pseudovirions and of authentic SARS-CoV-2 virus. Further, they show that SARS-CoV-2 ORF7a counteracts this inhibitory effect of SERINC5, at least in some cases by reducing its incorporation. They also present evidence suggesting that SERINC5 physically interacts with SARS-CoV-2 Spike and ORF7a. Unfortunately, the biological relevance of these observations is not entirely clear, because the effects of endogenous SERINC5 on SARS-CoV-2 infectivity were not examined.

Specific points:

1. A major concern is that the authors have examined the effects of SERINC5 only through overexpression in 293T cells. This approach typically yields SERINC5 levels that substantially exceed endogenous expression levels. Consequently, the antiviral effects of SERINC5, when expressed in 293T cells, tend to be considerably more pronounced than what is observed for SERINC5 expressed at endogenous levels. To demonstrate biological relevance, the authors need to show that endogenous SERINC5 in pneumocytes restricts SARS-CoV-2. This is not at all certain, since the endogenous SERINC5 expression levels in Fig. 7A appear rather low.
2. Fig. 5D: These are the only data in the paper that indicate that native ORF7a in its natural context can antagonize SERINC5 (codon-optimized ORF7a driven by the highly potent CMV promoter was used for the trans-complementation experiments). However, what is missing here is a crucial control showing that the replication of the deltaORF7a virus was not reduced in the absence of SERINC5.
3. Fig. 6: the evidence supporting the conclusion that ORF7a forms a complex with SERINC5 and SARS-CoV-2 S could be strengthened by including some controls. This is particularly important, because the authors chose to overexpress all the putative interaction partners together in 293T cells. This approach can easily lead to false positive results simply because of the high expression levels that are typically obtained. Indeed, all combinations that the authors tested in Fig. 6 yielded positive interactions. The data would be far more convincing if the authors could identify specific determinants that are required for the interactions. At a minimum, they need to show that irrelevant HA- or EGFP-tagged proteins that localize to the ERGIC do not interact with ORF7a.
4. Curiously, the ORF7a ectodomain, which constitutes the bulk of the protein, was found to be dispensable for counteracting SERINC5. This raises the question of whether the ORF7a ectodomain is also dispensable for the purported interactions with SERINC5 and with SARS-CoV-2 S. This

should be examined experimentally to determine whether the physical interactions shown in Fig. 6 are functionally relevant.

5. page.16, line 341: it should be explained why a codon-optimized ORF7a expression plasmid was used. Did ORF7a also antagonize SERINC5 when a non-codon-optimized version was inserted into the pCDNA-V5/His TOPO expression plasmid?

Reviewer #3 (Remarks to the Author):

The manuscript by Timilsina et al. studies the interaction of SARS-CoV-2 with the host factor SERINC5, a transmembrane protein involved in phospholipid biogenesis. SERINC5 is a restriction factor of retrovirus infection that is incorporated into virions, inhibiting virus infectivity.

The manuscript describes that SERINC5 also restricts SARS-CoV-2 entry and identifies 7a protein as a viral antagonist of SERINC5. These are novel and interesting results in the field of coronavirus-host interactions, with potential application for antiviral design.

However, there are some concerns regarding the experimental design and the conclusions of the study.

The work includes two different experimental approaches. The first one corresponds to two distinct pseudoviruses, defective HIV and MLV, expressing SARS-CoV-2 S protein. SERINC5 has been previously described to interact with Nef HIV and glycoGag MLV antagonist factors. From these experiments, it was concluded that the incorporation of SERINC5 in pseudoviruses reduced viral infectivity, measured as luciferase expression inside transduced cells, indicating that SERINC5 was inhibiting cell entry in a process mediated by SARS-CoV-2 S protein.

Further experiments with pseudoviruses showed that SERINC5 did not affect either S-ACE2 interaction or proteolytic activation of S protein by cathepsin-L. In contrast, SERINC5 inhibited membrane fusion.

Interestingly, viral transmembrane protein 7a was incorporated into pseudovirions, but it did not affect incorporation of SERINC5 into pseudovirions,

The second experimental approach consisted in true infections with SARS-CoV-2 and the deletion mutant SARS-CoV-2- Δ 7a, which represent a more physiological system. In contrast to results observed with pseudovirions, 7a protein was found to inhibit the incorporation of SERINC5 into SARS-CoV-2 virions. These results suggest that, in terms of 7a-SERINC5 interactions in membranes, pseudoviruses are not a good surrogate of an authentic SARS-CoV-2 infection and raise the concern about the validity of conclusions from experiments using pseudovirus.

Authors might consider to limit the manuscript to the work with infectious SARS-CoV-2, which represent a more physiological and natural experimental system, as it is hard to withdraw conclusions using pseudo-typed viruses.

Specific points

1. The effect on infectivity of SERINC5 incorporated into SARS-CoV-2 virions was evaluated by overexpressing the protein in infected cells. This raises the concern of the relevance of SERINC5 in a natural infection, when physiological levels of endogenous SERINC5 are expressed in infected cells. To answer this question, silencing experiments of SERINC5 in infected cells might be used.

2. Regarding the contribution of endogenous SERINC5 in natural infections, its expression in lung Calu-3 cells was studied by qPCR. It would be helpful to show the expression levels of the protein and SARS-CoV-2 titers.

3. In most of the western-blots in the manuscript, overexpressed SERINC5 is shown as a number of bands with different molecular weights. In some of them, the indicated band is not the most abundant and its mobility is variable (please compare blots in Figs. 2, 3, and 6). Sometimes, the band pattern also differs between virus and cell fractions (Fig. 2A). Please clarify these points.

4. The inhibitory effect of 7a protein on SARS-CoV-2 infectivity (Fig. 5D) was shown in infected cells overexpressing SERINC5. To confirm this effect in the context of a natural infection, SARS-CoV-2 growth should be analyzed in the absence of overexpressed SERINC5.

Response to Reviewer Comments

We would like to thank the reviewers for taking the time to provide comments for this paper.

Reviewer #1 (Remarks to the Author):

In the manuscript by Timilsina et al., the authors show that the protein SERINC5 is able to restrict SARS-CoV-2 infection in much the same way as it has been shown to restrict HIV-1 infection. They find this to be the case when using a SARS-CoV-2 Spike pseudotyped lentiviral reporter system, as well as the authentic SARS-CoV-2 virions. Furthermore, the authors attempt to illustrate a possible mechanism whereby SERINC5 might antagonize viral infection, while providing convincing data that the SARS-CoV-2 ORF7a counteract the action of SERINC5. The latter is perhaps the highlight of the paper. Overall, Timilsina et al. provide a thorough and well-designed study that sheds light on an as-yet uncharacterized virus-host interaction system by using a range of careful assays. However, in some cases the authors overstate their conclusions in a way that is not sufficiently supported by the data presented.

Major concerns

1. The authors do not clearly explain whether their statistical testing is performed on the raw or normalised values, such as in Fig. 1B, E and Fig. 3 and Fig. 5A, E where a specific treatment has been set to 100% and all other measurements are in relation to this. The authors indicate that they have used a parametric t-test – however, if the testing has been performed on the normalised data, this indicates a violation of the assumption of parametric testing (equal variance between groups) because normalisation in this way would have set the variation of the reference group to 0.

As requested, we have gone through the paper and we have modified the statistical tests performed in normalized data. We are now performing one sample t-test, a statistical hypothesis test used to determine whether an unknown population mean is different from a specific value, in normalized data as clearly mentioned in the revised figure legends.

2. The authors should make it clear that the detection of SERINC5 in western blots is made possible by expressing a SERINC5-FLAG fusion protein, which is then detected by an anti-FLAG antibody. In Fig. 2A and C, for example, this is not indicated but rather mentioned later in the Methods section. It is important to specify that the detected SERINC5 is not the authentic protein but a recombinant version.

We have now added information about the tags of all the proteins in the figures for all applicable western blots.

3. Where the authors have used the authentic SARS-CoV-2 virus, they have indicated infection levels as relative levels of S mRNA to GAPDH mRNA, such as Fig. 1C. This is an unusual way to represent SARS-CoV-2 infection. Instead, a more convincing and rigorous approach would have been to use a flow cytometric assay to assess how many cells are positive for the SARS-CoV-2 N protein, RT-qPCR for

relative copies of genomic RNA or plaque assays using the virions released into the supernatant. As it is, the data makes it difficult to estimate the level of viral infection in the system. A lack of change in SERINC5 levels, for example, could in fact be due to very low levels of infection.

As the reviewer advises, we have repeated the experiment in Fig 1c by RT-qPCR for relative copies of genomic RNA using primers that were previously used in another publication in this journal¹.

4. The authors should quantify their western blot data before claiming to see a change in the level of a protein. For example, in Fig. 5 E, the authors claim that the level of SERINC5 is reduced in the virions, yet they show only one representative western blot without replicates or quantitative data. Claims like these should be backed up with the appropriate replicated measurements and statistical testing.

We have now quantified the western blot in Fig 5b, c and e and provide that data in Fig 5.

5. Throughout the paper, the appearance of SERINC5 in the western blots displayed varies considerably. In Fig. 2A, it appears as two bands, in Fig. 5C as multiple different bands and Fig. 6B as a smear. The authors should address this in the text, at the very least, as this differing appearance makes interpretation of the results difficult.

The reason for the difference in appearance of SERINC5 across the western blots in the different experiments in this paper has to do with the fact that we are using different lysis buffers to lyse our cells or viruses in the different experiments (as described in the Materials and Methods section). Therefore, the differences in the processing of the samples prior to immunoblotting is responsible for the different patterns of SERINC5. Also, SERINC5 is a glycosylated protein², thus any changes in sample preparation affect the band patterns in western blots. We have addressed this issue in the Discussion section of the manuscript (line 442-444).

6. The authors claim that the antiviral action of SERINC5 is via antagonisation of S mediated fusion (line 246-247). Firstly, this assertion is too definitive given the data at hand. What would be needed is, for example, the same experiment (Fig. 4C) in the presence of a neutralising Spike antibody to illustrate that what is being inhibited is specifically S function. Furthermore, the authors should explain more clearly in the text as to why they decided to investigate cathepsin/TMPRSS2-mediated S cleavage and not furin-mediated cleavage (Fig 4B).

As the reviewer recommended, we have now repeated this experiment using a neutralizing antibody (BEI Resources NR55410) and as expected, addition of the neutralizing antibody abolished any signal from our Blam-Vpr assay; therefore the effect we are seeing is Spike-specific (see Fig 4c). We focused on the effect of SERINC5 on the Cathepsin/TMPRSS2-mediated S cleavage, because SERINC5 does not affect virus infection in the producer cells but in the target cells. Furthermore, cleavage of S by furin to S2 is unaffected by the presence or absence of SERINC5 as shown in Fig 2a. Therefore, we focused on the proteases acting on S in the target cells.

7. In Fig. 6A, the authors should provide a brightfield version of the microscopic images. Showing only the fluorescent channels makes it difficult to understand exactly what is being depicted, i.e. how many cells are in the image and where the cell membranes are located. Furthermore, the co-localisation of the proteins in the images should be quantified using the appropriate software to support the qualitative data further.

The reason we did not provide a bright version of the images in Fig 6A (now revised Fig 7A) is because the cells can be clearly seen in the images. In this image, at a high magnification we are showing two cells and the localization of the proteins of interest (SERINC5, Spike and ORF7a) and ERGIC53 is clearly depicted. We have taken images from a number of fields showing us similar results. As requested by the reviewer, to ensure SERINC5, ORF7A and SARS-CoV-2 Spike proteins colocalize at the ERGIC53, we performed colocalization analysis of a region of interest (ROI) defined by the presence of ERGIC53-GFP signal using the Coloc2 plugin from ImageJ (FIJI) from 3 different images we took, each from a different experiment. Spearman's rank correlation values and analysis (r_s , mean \pm SD) was used to determine co-localization and is shown below.

ERGIC53-SERINC5	ERGIC53-ORF7a	ERGIC53-Spike
0.669 \pm 0.070	0.693 \pm 0.095	0.814 \pm 0.075

The closer the r_s value is to ± 1 , the more correlated are 2 values (see <https://www.statsutor.ac.uk/resources/uploaded/spearman.pdf>), anything above 0.6 indicates a strong correlation. Our quantitative analysis above shows that SERINC5, ORF7a and Spike colocalize with ERGIC53, a marker of the ERGIC compartment.

Minor concerns

1. The authors refer vaguely to “databases of peripheral blood mononuclear cells (PBMCs)” (lines 101-102) but fail to explain what databases these are (transcriptomic?). Furthermore, the citation provided (numbered 19) does not make clearer what databases are being referenced. The authors should be more explicit in the text about this.

We agree with the reviewer, we have gone back and edited this part of the paper and included a new citation from a paper that specifically looked at the RNA levels of SERINC3 and 5. We are now explicitly stating that the data on ACE2 were derived from transcriptomic analyses (see line 96).

2. The authors should consider revising the grammar and language usage throughout the paper. For example, “we concluded that 159 SERINC5 gets incorporated in SARS-CoV-2 VLPs” (lines 158-159) is fairly informal, unacademic phrasing and “gets” should be replaced with “is”. The authors could reconsider the usage of the verb “get” throughout the paper.

We have now edited the document as advised by the reviewer.

3. SERINC5 is designated an “attractive target for the development of antiviral therapies” (line 461). This phrasing is confusing as antiviral therapies are targeted against a virus and not a host factor. Rephrasing and revision is advised.

We agree with the reviewer and have now edited this statement in the paper (line 491).

4. It should be made more clear in the main body of the text which virus strain has been used for which assay. This is discussed in the methods section (lines 576-601) but, worryingly, the authors fail to cite any papers or works outlining the origin of the different virus strains. Furthermore, it is generally advised to sequence the genomes of viral isolates via NGS to verify that the virus in use does not harbour any mutations that would indicate considerable variation from the original isolate. The furin cleavage site, for example, is often mutated in SARS-CoV-2 virions that have been propagated in cell culture. The authors do not provide this.

As advised by the reviewer, we have now included the strain of the virus (USA-WA1/2020) in the Results section in all infection experiments (line 104, 155 and 162). We also have added references in the Results and Methods section when describing the icSARS-CoV-2 WT and icSARS-CoV-2ΔOrf7a⁽³⁾ and when describing the USA-WA1/2020 strain⁽⁴⁾. Losing the furin cleavage site from our virus preparations was something we also worried as it has been shown to occur when cells grown in certain cell lines⁽⁵⁾. For this reason, all viruses used in our experiments were sequenced as previously done⁽⁵⁾, to verify that the furin cleavage site is intact (sequencing files can be provided upon request).

Reviewer #2 (Remarks to the Author):

HIV-1 Nef is known to enhance the infectivity of progeny virions by inhibiting the uptake of the host multipass transmembrane protein SERINC5 into virions. In the present study, the authors first show that several members of the SERINC family, including SERINC5, are expressed in a human lung cancer cell line. They then go on to show that SERINC5 can inhibit SARS-CoV-2 Spike-mediated entry by targeting virus-cell membrane fusion, both in the context of HIV-based pseudovirions and of authentic SARS-CoV-2 virus. Further, they show that SARS-CoV-2 ORF7a counteracts this inhibitory effect of SERINC5, at least in some cases by reducing its incorporation. They also present evidence suggesting that SERINC5 physically interacts with SARS-CoV-2 Spike and ORF7a. Unfortunately, the biological relevance of these observations is not entirely clear, because the effects of endogenous SERINC5 on SARS-CoV-2 infectivity were not examined.

Specific points:

1. A major concern is that the authors have examined the effects of SERINC5 only through overexpression in 293T cells. This approach typically yields SERINC5 levels that substantially exceed

endogenous expression levels. Consequently, the antiviral effects of SERINC5, when expressed in 293T cells, tend to be considerably more pronounced than what is observed for SERINC5 expressed at endogenous levels. To demonstrate biological relevance, the authors need to show that endogenous SERINC5 in pneumocytes restricts SARS-CoV-2. This is not at all certain, since the endogenous SERINC5 expression levels in Fig. 7A appear rather low.

We are now also providing SERINC expression levels in normal lung tissues collected from 3 adult donors and we see high levels of expression of all SERINC genes (except SERINC4) (Fig 1a). Therefore, these genes are expressed in tissues infected by SARS-CoV-2. To demonstrate biological relevance, we determined the effect of endogenous SERINC5 on SARS-CoV-2 infection in the following ways (see Fig 6):

- 1. To determine the effect of endogenous SERINC5 on SARS-CoV-2 entry and the role of ORF7a, we collected wild type and Δ ORF7a virus from either Calu-3 cells expressing SERINC5 (treated with siControl) or Calu-3, in which SERINC5 is knocked down (treated with siSERINC5). Viruses were collected and equal genome copies were used to infect 293T-hACE2. Six hours post infection, we found that viral RNA levels of the wild type SARS-CoV-2 (expresses ORF7a) were unaffected by the presence of SERINC5. On the other hand, viral RNA levels of Δ ORF7a virus produced in cells knocked down for SERINC5 were higher at 6 hours post infection when compared to viral RNA levels in cells infected with virus produced from cells treated with siControl (Fig 6a). Thus, endogenous SERINC5 reduces SARS-CoV-2 entry in the absence of ORF7a.*
- 2. We also examined the effect of SERINC5 depletion by shRNA, on the spread of SARS-CoV-2 WT and Δ ORF7a viruses. We found that in the case of the WT virus, virus replicated to similar extent in both SERINC5-expressing (shControl) and SERINC5-depleted cells (shSERINC5) by 48 hours post infection. On the other hand, the Δ ORF7a virus replicated more efficiently in the SERINC5-depleted (shSERINC5) cells compared to SERINC5-expressing cells (shControl) (Fig 6b-c). Therefore, we concluded that endogenous SERINC5 affects virus replication in the absence of ORF7a.*

2. Fig. 5D: These are the only data in the paper that indicate that native ORF7a in its natural context can antagonize SERINC5 (codon-optimized ORF7a driven by the highly potent CMV promoter was used for the trans-complementation experiments). However, what is missing here is a crucial control showing that the replication of the deltaORF7a virus was not reduced in the absence of SERINC5.

We are now including this as a supplemental figure showing that WT and Δ Orf7a replicate to similar levels in 293T-hACE2 cells (Extended data Fig 3b), which is in agreement with previous findings³

3. Fig. 6: the evidence supporting the conclusion that ORF7a forms a complex with SERINC5 and SARS-CoV-2 S could be strengthened by including some controls. This is particularly important, because the authors chose to overexpress all the putative interaction partners together in 293T cells. This approach can easily lead to false positive results simply because of the high expression levels that are typically obtained. Indeed, all combinations that the authors tested in Fig. 6 yielded positive interactions. The

data would be far more convincing if the authors could identify specific determinants that are required for the interactions. At a minimum, they need to show that irrelevant HA- or EGFP-tagged proteins that localize to the ERGIC do not interact with ORF7a.

We now present data that the transmembrane domain of ORF7a is responsible for the anti-SERINC5 effect of this SARS-CoV-2 accessory protein. We also provide colP data showing that mutating the transmembrane domain of ORF7a abolishes the interaction of ORF7a with SERINC5 and Spike (Fig 8c and d).

We also provide new data using TMED2, a transmembrane protein found at the ERGIC⁶, as negative control for our co-IPs. We show that TMED2 does not interact with SARS-CoV-2 ORF7a. Therefore, ORF7a interaction with SERINC5 and Spike is specific (Extended data Fig 5).

4. Curiously, the ORF7a ectodomain, which constitutes the bulk of the protein, was found to be dispensable for counteracting SERINC5. This raises the question of whether the ORF7a ectodomain is also dispensable for the purported interactions with SERINC5 and with SARS-CoV-2 S. This should be examined experimentally to determine whether the physical interactions shown in Fig. 6 are functionally relevant.

Please see the above comment regarding our experiment with the transmembrane domain mutant of ORF7a.

5. page.16, line 341: it should be explained why a codon-optimized ORF7a expression plasmid was used. Did ORF7a also antagonize SERINC5 when a non-codon-optimized version was inserted into the pCDNA-V5/His TOPO expression plasmid?

The reason we did our experiments with the codon optimized ORF7a is because ORF7a is expressed very poorly by itself in the absence of infection. As for non codon optimized ORF7a; that is what is present in our actual infection experiments (SARS-CoV-2 WT) and there we see that ORF7a counteracts SERINC5 (Fig 5d) just as we saw with our transfections. Therefore, our transfection data with codon-optimized ORF7a are mirrored by our data with ORF7a (non codon-optimized) during actual SARS-CoV-2 infection.

Reviewer #3 (Remarks to the Author):

The manuscript by Timilsina et al. studies the interaction of SARS-CoV-2 with the host factor SERINC5, a transmembrane protein involved in phospholipid biogenesis. SERINC5 is a restriction factor of retrovirus infection that is incorporated into virions, inhibiting virus infectivity.

The manuscript describes that SERINC5 also restricts SARS-CoV-2 entry and identifies 7a protein as a viral antagonist of SERINC5. These are novel and interesting results in the field of coronavirus-host

interactions, with potential application for antiviral design.

However, there are some concerns regarding the experimental design and the conclusions of the study.

The work includes two different experimental approaches. The first one corresponds to two distinct pseudoviruses, defective HIV and MLV, expressing SARS-CoV-2 S protein. SERINC5 has been previously described to interact with Nef HIV and glycoGag MLV antagonist factors. From these experiments, it was concluded that the incorporation of SERINC5 in pseudoviruses reduced viral infectivity, measured as luciferase expression inside transduced cells, indicating that SERINC5 was inhibiting cell entry in a process mediated by SARS-CoV-2 S protein.

Further experiments with pseudoviruses showed that SERINC5 did not affect either S-ACE2 interaction or proteolytic activation of S protein by cathepsin-L. In contrast, SERINC5 inhibited membrane fusion. Interestingly, viral transmembrane protein 7a was incorporated into pseudovirions, but it did not affect incorporation of SERINC5 into pseudovirions,

The second experimental approach consisted in true infections with SARS-CoV-2 and the deletion mutant SARS-CoV-2- Δ 7a, which represent a more physiological system. In contrast to results observed with pseudovirions, 7a protein was found to inhibit the incorporation of SERINC5 into SARS-CoV-2 virions. These results suggest that, in terms of 7a-SERINC5 interactions in membranes, pseudoviruses are not a good surrogate of an authentic SARS-CoV-2 infection and raise the concern about the validity of conclusions from experiments using pseudovirus.

Authors might consider to limit the manuscript to the work with infectious SARS-CoV-2, which represent a more physiological and natural experimental system, as it is hard to withdraw conclusions using pseudo-typed viruses.

Specific points

1. The effect on infectivity of SERINC5 incorporated into SARS-CoV-2 virions was evaluated by overexpressing the protein in infected cells. This raises the concern of the relevance of SERINC5 in a natural infection, when physiological levels of endogenous SERINC5 are expressed in infected cells. To answer this question, silencing experiments of SERINC5 in infected cells might be used.

We agree with the reviewer and we have used siRNAs and shRNAs to examine the endogenous effect of SERINC5. To demonstrate biological relevance, we determined the effect of endogenous SERINC5 on SARS-CoV-2 infection in the following ways:

- 1. To determine the effect of endogenous SERINC5 on SARS-CoV-2 entry and the role of ORF7a, we collected wild type and Δ ORF7a virus from either Calu-3 cells expressing SERINC5 (treated with siControl) or Calu-3, in which SERINC5 is knocked down (treated with siSERINC5). Virus was collected and equal genome copies were used to infect 293T-hACE2. Six hours post infection, we*

found that viral RNA levels of the wild type SARS-CoV-2 (expresses ORF7a) were unaffected by the presence of SERINC5. On the other hand, viral RNA levels of Δ ORF7a virus produced in cells knocked down for SERINC5 were higher at 6 hours post infection when compared to viral RNA levels in cells infected with virus produced from cells treated with siControl (Fig 6a). Thus, endogenous SERINC5 reduces SARS-CoV-2 entry in the absence of ORF7a.

- 2. We also examined the effect of SERINC5 depletion by shRNA, on the spread of SARS-CoV-2 WT and Δ ORF7a viruses. We found that in the case of the WT virus, virus replicated to similar extent in both SERINC5-expressing (shControl) and SERINC5-depleted cells (shSERINC5) by 48 hours post infection. On the other hand, the Δ ORF7a virus replicated more efficiently in the SERINC5-depleted (shSERINC5) cells compared to SERINC5-expressing cells (shControl) (Fig 6b-c). Therefore, we concluded that endogenous SERINC5 affects virus replication in the absence of ORF7a.*

2. Regarding the contribution of endogenous SERINC5 in natural infections, its expression in lung Calu-3 cells was studied by qPCR. It would be helpful to show the expression levels of the protein and SARS-CoV-2 titers.

We tried to look at the protein levels of SERINC genes in the Calu-3 cells but the antibodies that are commercially available gave us very variable and inconclusive results when we used them. Thus, we have not provided any protein data on SERINC genes in Calu-3 cells. As for titers, as recommended by reviewer 1 and is commonly seen in many reports, in Fig 1d we now show relative genomic RNA levels.

3. In most of the western-blot in the manuscript, overexpressed SERINC5 is shown as a number of bands with different molecular weights. In some of them, the indicated band is not the most abundant and its mobility is variable (please compare blots in Figs. 2, 3, and 6). Sometimes, the band pattern also differs between virus and cell fractions (Fig. 2A). Please clarify these points.

Please see response to reviewer 1 comment 5.

4. The inhibitory effect of 7a protein on SARS-CoV-2 infectivity (Fig. 5D) was shown in infected cells overexpressing SERINC5. To confirm this effect in the context of a natural infection, SARS-CoV-2 growth should be analyzed in the absence of overexpressed SERINC5.

Please see response to first comment by this reviewer.

References:

- Alexandersen S, Chamings A, Bhatta TR. SARS-CoV-2 genomic and subgenomic RNAs in diagnostic samples are not an indicator of active replication. *Nat Commun* **11**, 6059 (2020).

2. Sharma S, Lewinski MK, Guatelli J. An N-Glycosylated Form of SERINC5 Is Specifically Incorporated into HIV-1 Virions. *J Virol* **92**, (2018).
3. Hou YJ, *et al.* SARS-CoV-2 Reverse Genetics Reveals a Variable Infection Gradient in the Respiratory Tract. *Cell* **182**, 429-446 e414 (2020).
4. Harcourt J, *et al.* Severe Acute Respiratory Syndrome Coronavirus 2 from Patient with Coronavirus Disease, United States. *Emerg Infect Dis* **26**, 1266-1273 (2020).
5. Lamers MM, *et al.* Human airway cells prevent SARS-CoV-2 multibasic cleavage site cell culture adaptation. *Elife* **10**, (2021).
6. Jenne N, Frey K, Brugger B, Wieland FT. Oligomeric state and stoichiometry of p24 proteins in the early secretory pathway. *J Biol Chem* **277**, 46504-46511 (2002).

Reviewer comments, second round review –

Reviewer #2 (Remarks to the Author):

Overall, the authors have been quite responsive to the issues raised.

A major concern was whether endogenous SERINC5 levels in pneumocytes are sufficient to restrict SARS-CoV-2, especially since the endogenous SERINC5 levels in Calu-3 cells shown in Fig. 1b appear rather low. This concern is now alleviated by additional data indicating higher SERINC5 expression levels in normal lung tissue.

The authors now also show data in Fig. 6a suggesting that even the relatively low endogenous SERINC5 levels in Calu-3 cells reduce the infectivity of SARS-CoV-2 in the absence of ORF7A. Curiously, Fig. 6a also shows that the relative infectivity of the WT virus was lower when produced in Calu-3 cells depleted of SERINC5. The authors consider this negative effect insignificant, even so the positive effect of depleting SERINC5 on the virus lacking ORF7a was not much larger. In my opinion, since the biological relevance of the observations hinges to a large extent on this result, this experiment should be repeated to confirm that there is indeed a specific effect in the absence of ORF7A.

Reviewer #3 (Remarks to the Author):

The authors have fully addressed the reviewers points. I am suggesting just a minor complement, described in my letter to the authors and I believe that the manuscript will be ready for final acceptance.

Response to Reviewer Comments

We would like to thank the reviewers for taking the time to provide comments for our manuscript.

Reviewer #1 (Remarks to the Author):

The authors addressed most of my concerns in a largely satisfactory manner. However, I still advise to improve Figure 6A by either showing brightfield versions of the microscopic images because, contrarily to what the authors replied ("the cells can be clearly seen in the images") the border of the cells are invisible. A (second best) alternative to display brightfield images is to manually draw the surroundings of the shown cells.

Furthermore, the new information containing the calculations on the extent of colocalization of pairs of factors, which is provided in the response to the editors, must be included in the manuscript, which apparently does not seem to have happened yet.

As the reviewer advises, we have now shown cell borders and quantitative analyses of colocalization, which was previously in the "Response to reviewers" file, now has been added in Fig. 7a as the reviewer advised.

Reviewer #2 (Remarks to the Author):

Overall, the authors have been quite responsive to the issues raised.

A major concern was whether endogenous SERINC5 levels in pneumocytes are sufficient to restrict SARS-CoV-2, especially since the endogenous SERINC5 levels in Calu-3 cells shown in Fig. 1b appear rather low. This concern is now alleviated by additional data indicating higher SERINC5 expression levels in normal lung tissue.

We are happy that the reviewer liked this piece of data that we included in the previous revision. We agree with the reviewer that this is a very important piece of data.

The authors now also show data in Fig. 6a suggesting that even the relatively low endogenous SERINC5 levels in Calu-3 cells reduce the infectivity of SARS-CoV-2 in the absence of ORF7A. Curiously, Fig. 6a also shows that the relative infectivity of the WT virus was lower when produced in Calu-3 cells depleted of SERINC5. The authors consider this negative effect insignificant, even so the positive effect of depleting SERINC5 on the virus lacking ORF7a was not much larger. In my opinion, since the biological relevance of the observations hinges to a large extent on this result, this experiment should be repeated to confirm that there is indeed a specific effect in the absence of ORF7A.

We have now repeated the experiment depicted in Fig. 6a as the reviewer advised. As shown in Fig. 6a,

the relative infectivity of the WT virus is similar for viruses produced in Calu3 cells treated with siControl or siSERINC5 (depleted for SERINC5). Therefore the SERINC5 effect is specific to the absence of ORF7a.